# Strigolactone perception and deactivation by a hydrolase receptor DWARF14

Yoshiya Seto[1,2,3,4,5], Rei Yasui [1], Hiromu Kameoka [6,12], Muluneh Tamiru[7,13], Mengmeng Cao[1], Ryohei Terauchi[7,14], Akane Sakurada[1], Rena Hirano[1], Takaya Kisugi[1], Atsushi Hanada[1,2], Mikihisa Umehara[2,8], Eunjoo Seo[2], Kohki Akiyama[9], Jason Burke[3,4], Noriko Takeda-Kamiya[2], Weiqiang Li[2,15], Yoshinori Hirano[10], Toshio Hakoshima[10], Kiyoshi Mashiguchi [1], Joseph P. Noel [3,4], Junko Kyozuka[1,6] & Shinjiro Yamaguchi[1,2,11]

The perception mechanism for the strigolactone (SL) class of plant hormones has been a subject of debate because their receptor, DWARF14 (D14), is an α/β-hydrolase that can cleave SLs. Here we show via time-course analyses of SL binding and hydrolysis by *Arabidopsis thaliana* D14, that the level of uncleaved SL strongly correlates with the induction of the active signaling state. In addition, we show that an AtD14[D218A] catalytic mutant that lacks enzymatic activity is still able to complement the *atd14* mutant phenotype in an SL-dependent manner. We conclude that the intact SL molecules trigger the D14 active signaling state, and we also describe that D14 deactivates bioactive SLs by the hydrolytic degradation after signal transmission. Together, these results reveal that D14 is a dual-functional receptor, responsible for both the perception and deactivation of bioactive SLs.

[1] Department of Biomolecular Sciences, Graduate School of Life Sciences, Tohoku University, 2-1-1 Katahira, Aoba-ku, Sendai, Miyagi 980-8577, Japan. [2] RIKEN Plant Science Center, Tsurumi, Yokohama, Kanagawa 230-0045, Japan. [3] Jack H. Skirball Center for Chemical Biology and Proteomics, Salk Institute for Biological Studies, La Jolla, CA 92037, USA. [4] Howard Hughes Medical Institute, Salk Institute for Biological Studies, La Jolla, CA 92037, USA. [5] Department of Agricultural Chemistry, School of Agriculture, Meiji University, 1-1-1 Higashimita, Tama-ku, Kawasaki, Kanagawa 214-8571, Japan. [6] Graduate School of Agricultural and Life Science, University of Tokyo, Yayoi, Bunkyo, Tokyo 113-8657, Japan. [7] Iwate Biotechnology Research Center, 22-174-4 Narita, Kitakami, Iwate 024-0003, Japan. [8] Graduate School of Life Sciences, Toyo University, 1-1-1 Izumino, Itakura-machi, Ora-gun, Gunma 374-0193, Japan. [9] Graduates School of Life and Environmental Sciences, Osaka Prefecture University, 1-1 Gakuen-cho, Naka-ku, Sakai, Osaka 599-8531, Japan. [10] Structural Biology Laboratory, Nara Institute of Science and Technology, 8916-5 Takeyama, Ikoma, Nara 630-0192, Japan. [11] Institute for Chemical Research, Kyoto University, Gokasho, Uji, Kyoto 611-0011, Japan. [12] Present address: Graduates School of Life and Environmental Sciences, Osaka Prefecture University, 1-1 Gakuen-cho, Naka-ku, Sakai, Osaka 599-8531, Japan. [13] Present address: Department of Animal, Plant and Soil Sciences AgriBio, Centre for AgriBioscience, La Trobe University, 5 Ring Road Bundoora, Melbourne, VIC 3086, Australia. [14] Present address: Laboratory of Crop Evolution, Graduate School of Agriculture, Kyoto University, Mozume, Muko, Kyoto 617-0001, Japan. [15] Present address: Signaling Pathway Research Unit, RIKEN Center for Sustainable Resource Science, 1-7-22 Suehiro-cho, Tsurumi, Yokohama 230-0045, Japan. These authors contributed equally: Yoshiya Seto, Rei Yasui. Correspondence and requests for materials should be addressed to Y.S. (email: yoshiya@meiji.ac.jp) or to S.Y. (email: shinjiro@scl.kyoto-u.ac.jp)

Strigolactones (SLs) were originally characterized as root-derived signals for parasitic and symbiotic interactions[1,2], yet are now known as endogenous plant hormones that control shoot branching and diverse aspects of plant growth[3–7]. Canonical SLs have a four-ring structure (Fig. 1a), and recent studies have revealed how SLs are synthesized from carotenoid molecules via a key intermediate called carlactone (CL) (Fig. 1a, b)[8–10].

The SL receptor, DWARF14 (D14), was initially characterized from a rice SL-insensitive mutant, *d14*[11], and orthologues have since been identified from Arabidopsis (*Arabidopsis thaliana* D14; AtD14), petunia (DECREASED APICAL DOMINANCE2; DAD2), and pea (RAMOSUS3; RMS3)[12–14]. In this paper, the rice D14 is referred to as *Oryza sativa* D14 (OsD14) to distinguish from AtD14. The D14 enzymes belong to the α/β-fold hydrolase family and possess the canonical catalytic triad: Ser, His, and Asp. In fact, D14 acts as a hydrolase for some SLs, cleaving them into ABC-ring (ABC-formyltricyclic lactone; ABC-FTL), and D-ring (Hydroxymethylbutenolide; HMB) parts[8,12,13,15,16]. In rice, signal transduction occurs through an SL-dependent interaction between OsD14 and a negative regulator of SL signaling, D53 (SUPPRESSOR OF MAX2 1-LIKE (SMXL) 6/7/8 in Arabidopsis), leading to the rapid degradation of D53 through the 26 S proteasome pathway in a manner that requires the F-box protein, D3 (MORE AXILLARY GROWTH2 (MAX2) in Arabidopsis)[17–20] (Fig. 1b). The receptor function of D14 in the SL pathway has been established; however, the role of SL hydrolysis in the signaling mechanism has been a subject of debate. A significant question is when and how the signal is transduced while D14 interacts with SLs: is the signal transduced before, during, or after the hydrolysis of SLs by D14?

Recently, two groups proposed a model in which the hormone signaling is mediated by a covalently linked reaction intermediate of SL with D14[13,16]. A covalent modification involving His of D14 catalytic triad with the D-ring part of SL derived molecule was detected by mass spectrometry analysis. Moreover, Yao et al. solved a crystal structure of an AtD14-D3-ASK1 multiprotein signaling complex induced by an SL analog, GR24 (Fig. 1a)[16]. The authors proposed that in this signaling complex AtD14 possesses a D-ring derived molecule, which was covalently linked with AtD14 bridging the Ser with the His of the catalytic triad. The authors named this sealed molecule to be covalently linked intermediate molecule (CLIM). Moreover, significant conformational changes were observed for AtD14 in the multiprotein complex when compared with its apo structure. Based on these results, a model has been proposed in which the CLIM is necessary for the induction of conformational changes in the AtD14 protein and the SL signal is transduced during its hydrolysis. However, very recently Carlsson et al. reanalyzed this complex structure data and found that the electron density in the active site is too small to accommodate the proposed CLIM[21]. More likely, the substance bound to the active site is not the CLIM, rather an iodine ion from the crystallization reagents[21]. In addition to the uncertain structure data, the hydrolysis reaction by D14 was reported to be extremely slow, on the order of hours[12,15], whereas degradation of the target protein, D53/SMXLs, was detected within 5–20 min after the SL treatment in planta[17–20]. Thus, the CLIM model, which requires the hydrolysis reaction of D14 to transmit the signal, is inconsistent with this rapid response. Taken together, although the structural characterization of the D3-bound form of AtD14 was a significant breakthrough in this research field, the signaling mechanism, in particular the chemical signal that induces the active signaling state of D14, remains a matter of debate.

Here, we report that the active signaling state of D14 can be triggered upon intact SL binding, but not by the hydrolysis intermediate. We also demonstrate that D14 can deactivate SLs by hydrolytic degradation after signal transmission.

## Results

**Structural requirements for the D14-SL interaction.** In order to understand the relationship between the hydrolysis reaction and the signal transducing role of D14, we first comprehensively examined the structural requirements for the D14-SL interaction using various naturally occurring SLs and synthetic analogs by hydrolysis assays and differential scanning fluorimetry (DSF) experiments, which can evaluate protein-chemicals interactions by monitoring protein melting temperature (Tm) shifts induced upon exposure to chemicals. The hydrolysis kinetics of the AtD14-catalyzed hydrolysis reaction were measured using the naturally occurring single isomer of 5-deoxystrigol (5DS) (Fig. 1a). We detected two previously reported hydrolysis products, ABC-FTL and HMB, generated in a catalytic triad Ser-dependent manner (Supplementary Fig. 1a, b)[12], and the $K_m$, $V_{max}$, and $K_{cat}$ were calculated to be 4.9 μM, 4.0 nmol/min/mg protein, and 0.12 min$^{-1}$, respectively (Supplementary Fig. 1c). Among the stereoisomers of SLs, (2′R)-isomers have a demonstrably greater effect on the inhibition of shoot branching than the (2′S)-isomers in both Arabidopsis and rice[22,23] (Supplementary Fig. 2a). We observed that AtD14 hydrolyzes the (2′R)-isomers more efficiently than the (2′S)-isomers across all tested SLs (Supplementary Fig. 2b). OsD14 also preferred (2′R)-isomers of 5DS (Supplementary Fig. 2c). A biologically inactive analog, in which the double bond of the enol ether bridge is replaced by a single bond, 3,6′-dihydroGR24[22], was not hydrolyzed by AtD14/OsD14 (Supplementary Fig. 2d). Debranones, such as Br-PMF and CN-PMF, are a class of non-enol ether-type SL analogs (Supplementary Fig. 2e). They were reported to inhibit shoot branching in an AtD14/OsD14- and MAX2/D3- dependent manner[23–25]. We quantitatively examined the hydrolyzability of Br-PMF and CN-PMF by AtD14, which demonstrated that both of them are significantly poorer substrates when compared with GR24 (racemic mixture, Supplementary Fig. 2e). The low catalytic activity of AtD14 for debranones, coupled with the observations that debranones induce a similar signaling response to GR24 *in planta*[23–25], raises the question of whether SL hydrolysis is required for D14-mediated SL signaling.

We next performed DSF experiments using various SLs and analogs. Previously, bioactive SLs were found to lower the Tm of D14[8,12,13,26], including GR24 and a newly found endogenous SL-like molecule called methyl carlactonoate (MeCLA), whereas the SL biosynthetic precursors such as carlactone (CL) or its carboxylated derivative, carlactonoic acid (CLA), were not able to induce clear temperature shift of AtD14 (Fig. 1a)[8]. An AB-ring truncated analog, GR5 (Supplementary Fig. 2a), was reported to inhibit shoot branching, and the (2′R)-isomer, (+)-GR5, showed much stronger activity than the (2′S)-isomer in shoot branching inhibition[22]. Here we found that only (+)-GR5, but not (−)-GR5, induces a clear melting temperature shift similar to 5DS (Fig. 2a, Supplementary Fig. 2f). By comparison, the biologically inactive analogs, such as ABC-FTL, HMB, and 3,6′-dihydroGR24, were insufficient to induce the same Tm shifts (Fig. 2a). Debranones were also reported to induce a Tm shift for OsD14/DAD2[26], and here we show that CN-PMF clearly induces a Tm shift for AtD14/OsD14 similar to 5DS, despite the fact that CN-PMF is less hydrolyzable (Fig. 2a and Supplementary Fig. 2f). Taken together with the hydrolysis assays, these data conclusively demonstrate that the Tm shift of D14 directly correlates with biological activity of the SL-related compounds, suggesting that the temperature shifts reflects the induction of an active signaling state of D14, possibly related to the conformational changes seen in the

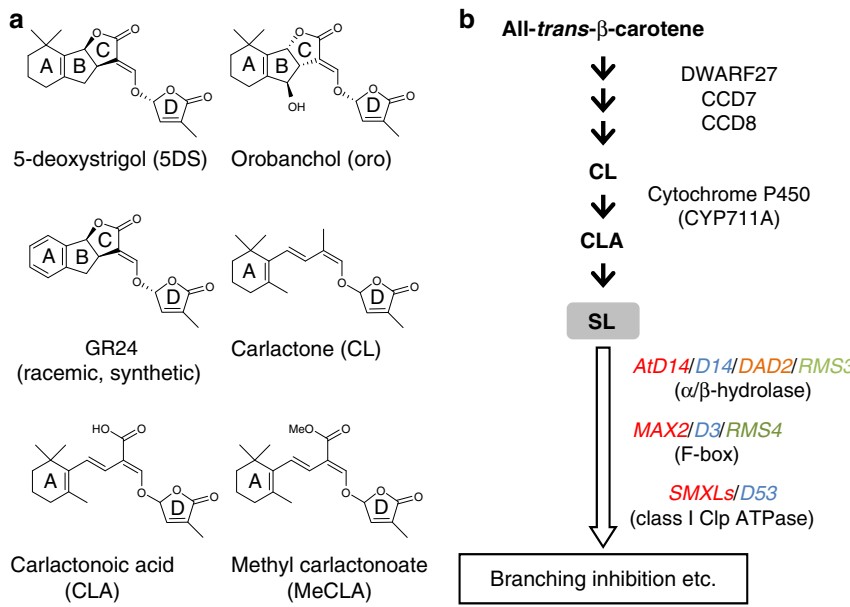

**Fig. 1** Chemical structures of SL-related compounds and a scheme for their biosynthesis and signaling pathways. **a** Structures of SL-related compounds. **b** The scheme for the SL biosynthesis and signaling pathways. Red, blue, orange, and green characters indicate genes of Arabidopsis, rice, petunia, and pea, respectively. Black arrows indicate the biosynthetic steps, and a white arrow indicates the signaling step. (CCD; carotenoid cleavage dioxygenase)

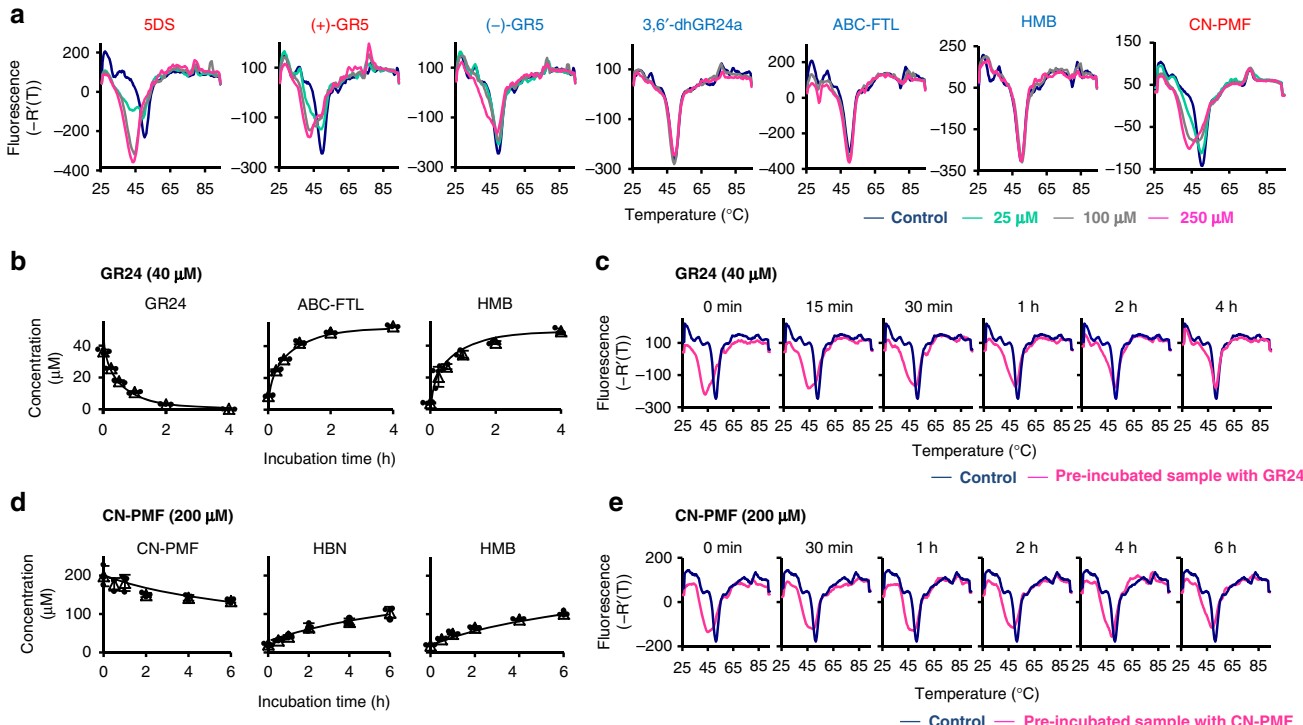

**Fig. 2** Evaluation of the AtD14-SL interaction using hydrolysis and DSF assays. **a** Melting temperature curves of AtD14 in the presence of various SLs and analogs. The names in red and blue denote biologically active and inactive (or weakly active) compounds, respectively. **b** and **d**, Monitoring by LC-MS/MS of the AtD14 hydrolysis reaction of GR24 (**b**) and CN-PMF (**d**). HBN; Hydroxybenzenenitrile. Data are the means ± SD ($n = 3$). **c** and **e** Melting temperature curves of AtD14 pre-incubated with GR24 (**c**) and CN-PMF (**e**) for indicated time period. Source data are provided as a Source Data file

AtD14-D3-ASK1 complex structure. In addition, a previous report demonstrated that the biologically active SL-related compounds could specifically promote OsD14-D3 interaction[26]. Given that the bioactive SL-dependent Tm shift observed in our DSF experiments, thermal destabilization of OsD14 is likely correlated with D3 binding.

**Time course analysis of D14 activation and SL hydrolysis.** In order to determine when the active state of D14 is triggered relative to the timing of SL hydrolysis, we designed a time-course DSF experiment coupled with quantitative detection of GR24 and its hydrolysis products. RMS3 was proposed to be a single turnover enzyme with a pro-fluorescent analog, GC242[13]. The authors

suggested the hydrolyzed D-ring was trapped through a covalent bond with the catalytic triad His, and this might be the reason for the single turnover reaction. However, we observed that all of the GR24 was consumed when the reaction was performed at a 1:6 molar ratio of AtD14:GR24 (Fig. 2b). This clearly demonstrates that AtD14 is not a single turnover enzyme with GR24. Interestingly, we further observed that ABC-FTL and HMB were released at almost same rate that GR24 was consumed (Fig. 2b). Generally, the hydrolysis reaction catalyzed by this protein family is a two-step process[27]. First, the activated hydroxyl group of the catalytic triad Ser acts as a nucleophile attacking to possibly the D-ring part of SLs to release the first product, ABC-FTL. At the same time, the protein forms a covalently-linked reaction intermediate with the D-ring part through the Ser residue of D14. Then, if we apply a conventional reaction model of this protein family[27], the activated water molecule triggers the second attack to the D14/D-ring complex to cleave the covalent bond, and HMB is released as the second product. On the other hand, according to two recent reports[13,16], the D-ring part is likely transferred to the catalytic triad His residue before its release as HMB, which was speculated to be significantly slower. However, our observation makes it unlikely that the D-ring derived product gets covalently trapped within the protein for a significant amount of time because the HMB release was detected at almost same turnover as the ABC-FTL release, and suggests that the second attack from a water molecule must occur immediately after the first attack from the catalytic Ser. Even if the covalently linked modification of His takes place, it must be released quickly as a product. Thus, it is likely that the first attack is rate limiting, which may cause slow catalysis.

We then performed the DSF experiments simultaneously with hydrolysis monitoring in order to observe the correlation between the degree of hydrolysis and the Tm shift. Strikingly, the maximum Tm shift was detected upon initial incubation with SL (0 min), before gradually returning to the control Tm over the course of the hydrolysis reaction (Fig. 2c), such that the GR24 consumption curve was highly correlated with the degree of the Tm shift (Supplementary Fig. 3a, b). These results strongly suggest that the signal that induces the Tm shift of AtD14 is GR24, is not the hydrolysis intermediate or products, suggesting that the active state of AtD14 is triggered upon GR24 binding, prior to its hydrolysis.

When GR24 was used as a substrate, we found that the background hydrolysis reaction was ongoing during DSF detection (Supplementary Fig. 3c). To observe the correlation between the hydrolysis reaction and the temperature shift more precisely, we used a less-hydrolyzable analog, CN-PMF. As expected, CN-PMF was much more stable than GR24 during DSF detection (Supplementary Fig. 3d). Although, the effect of CN-PMF on the Tm shift of AtD14 was smaller when compared with GR24, the slight shift persisted for a longer time because the rate of CN-PMF hydrolysis was much slower than GR24 (Supplementary Fig. 3e, f). To monitor the Tm shift more closely, the concentration of CN-PMF was increased and we were able to observe a clear correlation between substrate consumption and the Tm shift that is similar to GR24 (Fig. 2d, e and Supplementary Fig. 3g-i). Taken together, our data suggest that the uncleaved substrate itself, not the hydrolysis intermediate or the products, induces the Tm shift of AtD14. In contrast to reports regarding the role of CLIM in inducing conformational changes of AtD14, our results strongly suggest that the active state of AtD14 is triggered by the intact SL molecule prior to its hydrolytic degradation. Given that the two products were released at almost the same turnover rate, the lifetime of the covalently attached intermediate must be exceedingly short. It is unlikely that such a short-lived intermediate induces the drastic conformational changes necessary to recruit other partner proteins[16].

To address this question more directly, we analyzed the D-ring modification of AtD14 using LC-MS/MS during hydrolysis. Consistent with the reported data[13,16], we also detected a peak whose molecular mass increased by 98 ± 5, suggesting that AtD14 forms a covalently attached intermediate with the D-ring part of SL during the hydrolysis reaction (Supplementary Fig.4a). This conjugate peak was detected in both the native and denatured samples (Supplementary Fig. 4a). We performed a time-course analysis of this modified AtD14, and found that the intensity of the conjugate peak initially increased from 0 to 15 min, then gradually decreased over-time until 4 h when most of the AtD14 protein existed as unbound form. These results suggest that the D-ring part, which is once bound to AtD14, is released rapidly without being trapped tightly (Supplementary Fig. 4b). Moreover, it is noteworthy that the level of the covalently-linked AtD14 does not correlate with the degree of the Tm shift of AtD14, suggesting that the reaction intermediate is not a chemical that induces the transition of the AtD14 state. Because a small peak for modified AtD14 was still detectable after 4 h when all the substrate, GR24, was consumed, we individually incubated AtD14 with a reaction product, HMB. We found that even HMB alone could induce the formation of modified AtD14 at 400 μM (Supplementary Fig. 4c). Although at a lower concentration (40 μM), the modified AtD14 peak was almost undetectable, HMB produced in the ligand binding pocket by the hydrolysis reaction possibly forms the conjugate more effectively than the exogenously applied HMB, because HMB was reported to be trapped in the pocket after its release as a product[28]. Thus it would be difficult to distinguish whether this modified protein is a hydrolysis intermediate or a conjugate formed with HMB after the completion of hydrolysis. We analyzed the covalently modified AtD14 formation using CN-PMF as a substrate, and found a tendency similar to the case with GR24 (Supplementary Fig. 4d and e). Notably, a gradual increment of the modified AtD14 peak was more clearly observed in the case of CN-PMF than GR24, which does not correlate with the melting temperature shift. These results again support the idea that the substrate, but not the reaction intermediate, induced the melting temperature shift.

Our time-course DSF experiments demonstrate that the Tm of AtD14 is initially lowered by SL binding before it returns to the unbound state temperature. This result indicates that the SL-inducible dynamics of AtD14 is reversible. To further examine this, we spiked in fresh GR24 to the 4 h pre-incubated sample in which all the initially added substrate was consumed. As a result, the enzyme was still capable of hydrolyzing freshly added GR24, and fresh GR24 could induce the melting temperature shift of pre-incubated AtD14. These results again demonstrate that AtD14 is not a single turn over enzyme with GR24, and that the SL-inducible transition of AtD14 state is reversible depending on the presence of intact SL (Supplementary Fig. 5). Moreover, our results clearly reveal that debranones, which are much less hydrolyzable than GR24, cause the Tm shift to last for a longer time, thus providing an explanation for the potent biological activities of these analogs.

**Functional analysis of D14 catalytic triad mutants.** To evaluate the necessity of the hydrolytic function of D14, we next prepared four catalytic triad mutants (AtD14[S97A], AtD14[S97C], AtD14[D218A], and AtD14[H247A]) and found that the hydrolase activities of all of these mutants are drastically reduced when compared with a negative control (Fig. 3a). Although the RMS3[S96C] mutant was reported to have slight hydrolysis activity for GC242[13], the hydrolysis activity for 5DS in the corresponding mutant in Arabidopsis, AtD14[S97C], was reduced to the same level as other catalytic triad mutants (Fig. 3a). For each

mutant we also observed that all of the substrate stayed intact after the incubation with enzyme, suggesting that the hydrolase reaction gets stuck before the initial nucleophilic attack step (Supplementary Fig. 6a). To investigate the signal transducing function of these mutants, complementation tests were performed by expressing each mutant in the Arabidopsis atd14-2 mutant background under the control of the cauliflower mosaic virus (CaMV) 35 S promoter. The expression of AtD14$^{S97A}$ and AtD14$^{H247A}$ did not complement the atd14 mutant, which corroborates previous reports (Fig. 3b, c, and Supplementary Fig. 6b)[12,13]. Very interestingly, we found that AtD14$^{D218A}$, which has not previously been tested, completely complemented the atd14 mutant phenotype (Fig. 3b, c and Supplementary Fig. 6b). In a previous report, AtD14$^{S97C}$ was unable to complement the atd14 mutant when it was expressed as 6 × HA tag fusion[13]; however, we observed that the expression of untagged AtD14$^{S97C}$ partially complemented the atd14 mutant phenotype (Fig. 3b, c and Supplementary Fig. 6b). In addition, we found that AtD14$^{D218A}$ interacted with SMXL7 in an SL-dependent manner in yeast two hybrid (Y2H) experiments, suggesting that AtD14$^{D218A}$ is still capable of signal transduction despite lacking hydrolase activity, while AtD14$^{S97C}$ weakly interacts with SMXL7 in the presence of MeCLA (Fig. 3d and Supplementary Fig. 6c). We also found that AtD14$^{D218A}$ interacted with another signaling partner, MAX2, as examined by Y3H experiments, in which an SCF complex component, ASK1, was co-expressed without any tag as a third protein (Supplementary Fig. 6d). Interestingly, the SL-dependent interaction between AtD14$^{D218A}$ and MAX2 was observed only in the presence of ASK1, possibly because ASK1 stabilizes MAX2. The DSF experiments using the catalytic triad mutant proteins revealed that 5DS lowered Tms for AtD14$^{S97C}$ and AtD14$^{D218A}$ slightly, but did not change the Tms for AtD14$^{S97A}$ or AtD14$^{H247A}$, consistent with the observation that AtD14$^{S97C}$ and AtD14$^{D218A}$ retain the capacity for SL signaling in planta (Supplementary Fig. 7a). According to the D14 apo structures, the catalytic triad Ser and His are present at the surface of the active site pocket, whereas the Asp residue does not form part of the pocket surface[12,15,28,29] (Supplementary Fig. 7b), suggesting that these two residues might be important not only for the catalytic triad formation but also for the direct interaction with the ligand/substrate molecules. Therefore, the mutation to Ser and His possibly affected the initial interaction with SLs. Consistent with this idea, we found that AtD14$^{D218A}$ and AtD14$^{S97C}$ mutant proteins, which were capable of signal transduction, exhibit higher binding activity with 5DS than other two mutants, AtD14$^{S97A}$ and AtD14$^{H247A}$ (Supplementary Fig. 7c). Because the Tm curve of AtD14$^{D218A}$ suggested that this protein was unstable, even in the absence of 5DS (Supplementary Fig. 7a), we generated transgenic plants expressing AtD14$^{D218A}$ in the max4 atd14 double knockout mutant background (max4; CCD8 knockout defective for SL biosynthesis) to examine the SL dependency of this mutant protein function. These transgenic lines exhibited the severe branching phenotype, which was rescued by GR24 treatment, demonstrating that AtD14$^{D218A}$ complements the atd14 mutant in an SL-dependent manner (Fig. 3e, Supplementary Fig. 8a). Furthermore, in this background we observed that AtD14$^{D218A}$ expressing plants were more sensitive to GR24 than the AtD14$^{WT}$ expressing plants, possibly because of the defect in the hydrolytic degradation of GR24 by AtD14$^{D218A}$ (Fig. 3e), implying that the hydrolytic degradation of SL by D14 would be a deactivating step of bioactive SL. We also examined the function of the corresponding mutant in rice, OsD14$^{D268A}$, and found that it also complemented the d14 mutant phenotype despite lacking hydrolase activity (Supplementary Fig. 8b-e). These results, together with the time-course DSF experiments,

strongly suggest that the hydrolase reaction catalyzed by D14 is not necessary for the signal transducing role.

**Reanalysis of the conformationally altered AtD14 structure.** Conformational changes in AtD14, as shown in the complex with D3, were observed around the helical lid domain, through the rearrangement of α-helix structures, and as an open-to-close transition of the V-shaped lid domain[16]. In addition, the geometry of the catalytic triad is disrupted in this structure due to the reordering of the loop containing the catalytic Asp (D218) residue (Asp loop)[16]. The observed changes were reported to cause a significant reduction in the size of the active site pocket due to closure of the V-shaped lid (Supplementary Fig. 9a), and thus the authors suggested that the structural changes were triggered by a small molecule such as CLIM. On the other hand, our aforementioned data strongly suggest that the D14 active state is triggered by intact SL molecules. We downloaded the structural data of AtD14-D3-ASK1 complex (PDB code; 5HZG), and reanalyzed the structural changes in AtD14. Intriguingly, we found that the conformational change of the D218-containing loop enlarges the cavity adjacent to the catalytic Ser. The calculated volume of this new pocket, 944 Å$^3$, is larger than that of the original pocket in the apo structure, 863 Å$^3$ (Supplementary Fig. 9a). Molecular docking experiments using the structurally altered AtD14 structure showed favorable binding of GR24 and CN-PMF within the newly formed pocket, suggesting that AtD14 is able to accommodate an entire SL molecule after its conformational changes (Supplementary Fig. 9b). These results support the idea that the intact SL molecules, but not the hydrolysis intermediate, induce the conformational changes to D14.

**Functional analysis of a new d14 allele.** Our data suggests that the hydrolysis function of D14 is not necessary for its role in signal transduction. What then is the physiological role of SL hydrolysis by D14? To address this question, we used a mutant version of D14 characterized from a phenotypic screen for shoot branching in rice. We characterize a rice new d14 allele, d14-2 (osd14-2 in this paper to distinguish from atd14-2), containing a missense mutation of a highly conserved amino acid, R233H (Fig. 4a, Supplementary Fig. 10). Interestingly, both OsD14$^{R233H}$ and the corresponding mutant in Arabidopsis, AtD14$^{R183H}$, showed hydrolase activity similar to each wild type (Fig. 4b). However, AtD14$^{R183H}$ neither complemented the atd14 mutant phenotype (Supplementary Fig. 11a-c), nor interacted with SMXL7 in the presence of SLs (Supplementary Fig. 11d). Thus this mutation significantly influenced the signal transducing role of D14 without affecting its hydrolase ability, suggesting that these mutant proteins would be a good tool to further investigate the physiological role of SL hydrolysis by D14. As mentioned above, the results with AtD14$^{D218A}$ mutant implied that the hydrolysis function of D14 is a deactivating step of bioactive hormone compounds because the transgenic plants expressing this enzymatically inactive mutant in the atd14 max4 double mutant was highly sensitive to exogenously applied SL. If we hypothesize correctly, overexpression of the osd14-2 type mutant protein, which has only the hydrolase function, should cause the SL-deficient phenotype due to the reduction in bioactive hormone levels as were the cases with other plant hormones deactivating enzymes[30]. Based on this speculation, we overexpressed OsD14$^{R233H}$/AtD14$^{R183H}$ in each WT background (Nipponbare/Col-0) under the control of the CaMV 35 S promoter, which resulted in increased branching phenotype (Fig. 4c, d and Supplementary Fig. 11e-g). We also found that the levels of an endogenous SL, 4-deoxyorobanchol (4DO), in the rice overexpressors were decreased relative to WT and the empty vector-expressing plants (Fig. 4e). The SL hydrolysis products

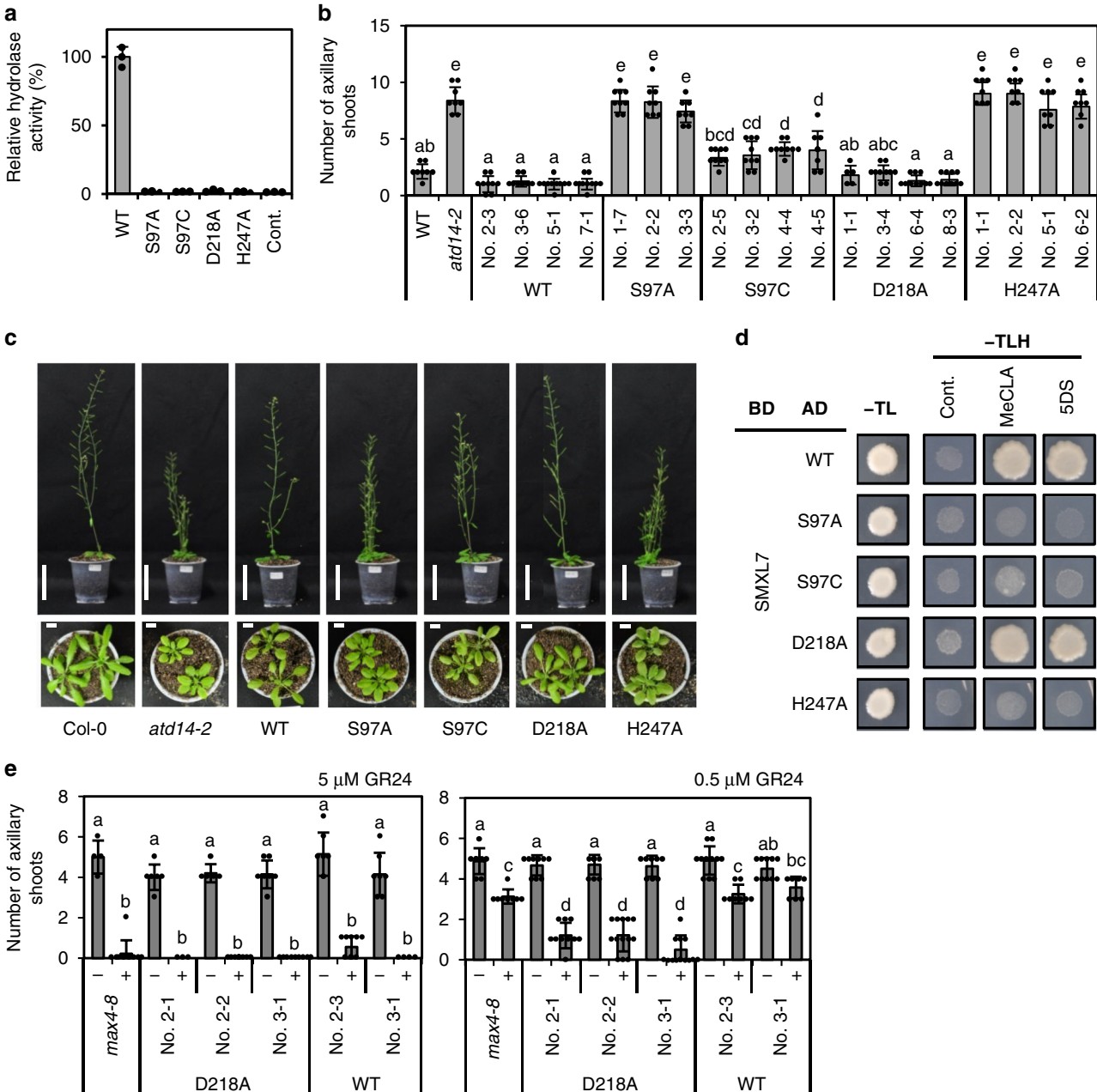

**Fig. 3** In vitro and in vivo functional analysis of catalytic triad mutants of AtD14. **a** Hydrolysis activities of catalytic triad mutants of AtD14 using 1 μM of 5DS as a substrate. Data are the means ± SD ($n = 3$). The control reaction (Cont.) is for MBP protein only. **b** No. of axillary shoots (over 5 mm) of Arabidopsis transgenic plants expressing each catalytic triad mutant of AtD14 in the *atd14-2* mutant background. Data are the means ± SD ($n = 5$–10, Different letters indicate significant differences at $P < 0.05$ with Tukey-kramer multiple comparison test.). **c** Phenotypes of Arabidopsis transgenic plants expressing each catalytic triad mutant of AtD14. Mature 50 days old plants phenotypes (upper panel) and leaf morphology phenotypes of 25 days old plants (lower panel) are shown. Scale bars = 5 cm (upper panel), 1 cm (lower panel). **d** Y2H analysis of the interaction between SMXL7 and each catalytic triad mutant of AtD14. Yeast transformants were spotted onto the control medium (SD−Leu/−Trp (−TL)) and selective medium (SD−Leu/−Trp/−His (−TLH)) in the absence or presence of SLs (10 μM *rac*-MeCLA or 10 μM 5DS). Control (Cont.) is acetone only. **e** Shoot branching inhibition assays of the Arabidopsis transgenic lines expressing AtD14^WT and AtD14^D218A, respectively, in the *atd14 max4* double mutants background. The bars indicate the No. of axillary shoots (over 5 mm) in the presence (+) or absence (−) of GR24 at 5 μM (left panel) and 0.5 μM (right panel), respectively. Data are the means ± SD ($n = 3$–13, Different letters indicate significant differences at $P < 0.05$ with Tukey-kramer multiple comparison test). Source data are provided as a Source Data file

were reported to have quite weak or no biological activities in shoot branching inhibition[12,28]. Moreover, our time-course analysis revealed that the signal is transduced prior to SL hydrolysis. Considering all together, our data support the hypothesis that the hydrolase reaction catalyzed by D14 would be a deactivating step of SLs after transducing the signal.

## Discussion

We conclude that intact SL molecules induce the active signaling state of D14, and that D14 deactivates bioactive SLs by the hydrolytic degradation after signal transmission. Therefore, our data demonstrate that D14 is a dually functional protein (Fig. 5). Notably, we could successfully separate two functions of D14

biochemically and genetically just by introduction of single amino acid substitution.

Our data enable us to speculate more detailed mechanisms of signal transduction as illustrated in Fig. 5. When a bioactive SL binds the active site pocket of D14, it induces conformational changes to D14[16]. Considering the presence of the enlarged pocket observed in the AtD14-D3 protein complex, it seems likely that SL initially induces a conformational change of the loop containing the catalytic triad Asp. After the formation of the enlarged pocket, SL may translocate within this pocket, triggering the open-to-close transition of the helical lid domain. Although there has been no evidence of such detailed events, our time-course DSF experiments together with the catalytic triad mutant analysis demonstrate that the induction of the D14 active state is triggered by an intact SL molecule, not by the hydrolysis intermediate or products. This means that the signaling process does not require the hydrolytic function of D14. In the AtD14-D3-ASK1 complex structure, an intact SL molecule was not observed in this newly formed pocket. It is likely that SL in the AtD14-D3-ASK1 complex crystal was partly hydrolyzed during crystallographic experiments due to a relatively high pH condition[16], resulting in poor electron density for the ligand molecules. Or it is also possible that the disordering of the Asp loop induced the detachment of the SL molecule during crystallization or other experimental procedure.

Upon binding of the intact SL, D14 initially adopts a destabilized conformation characteristic of a catalytically inactive state that is due to the disruption of the catalytic triad formation. In this state, the conformationally altered D14 protein interacts with its signaling partners, D53/SMXLs and D3/MAX2, to transmit the SL signal. As seen in the AtD14-D3-ASK1 complex structure, D3/MAX2 F-box interacts with the activated D14 protein at the surface of the rearranged helical lid domain. In this complex, D53/SMXLs may bind around the Asp loop region. In our model, after degradation of the negative regulator of SL signaling, the loop returns to reconstruct the catalytic triad, which then induces the hydrolytic degradation and deactivation of bioactive SLs (Fig. 5). The slow rate of the first attack from the Ser residue, as suggested by our hydrolysis monitoring, can be explained by this temporary physical disruption of the catalytic triad. The activation of hormone degradation pathways in signaling is exemplified in other plant hormone pathways, such as GA, and is considered to be an important mechanism to maintain homeostatic levels of active hormones in planta[30]. Similarly, our data suggest that the D14 receptor protein in the SL signaling pathway is responsible for SL deactivation by hydrolytic catalytic activity native to the α/β-fold hydrolase family.

In conclusion, our targeted biochemical and genetic experiments uncover previously undetected, yet critical aspects of the signaling mechanisms of SLs, as they are mediated by a catalytically active α/β-hydrolase family protein that is responsible for both the perception and deactivation of bioactive hormone signals.

## Methods

**Plant materials and growth conditions.** We used rice cultivar (*Oryza sativa* L. cv. Nipponbare) as the WT. The rice *d14-1* mutant was used after backcrossing with Nipponbare 3 times (*d14-1N*) for construction of transgenic plants[11]. The rice *osd14-2* was characterized from Sasanishiki EMS mutant lines. We used Arabidopsis ecotype Col-0 as the WT, *max4-8* (SALK_072750)[3], and *atd14-2* mutants[9]. For Arabidopsis phenotype observation, the seeds were directly sown on soil and grown under the long day condition (16 h light/8 h dark) at 22 °C. For shoot branching assays, a hydroponic culture system was used; the growth condition is described below. For the rice phenotype observation, the plants were grown under long day conditions (16 h light at 28 °C/8 h dark at 25 °C). The number of tillers was measured after 42 days of growth. For SL measurements in rice, the hydroponic culture system was used.

**Chemicals.** Br-PMF and CN-PMF were purchased from Chiralix. Other SLs were prepared as part of our previous studies[22,31].

**Functional expression of AtD14 and OsD14 proteins.** The coding sequences for OsD14 and AtD14 were amplified by PCR from cDNA synthesized from total mRNA of the rice and Arabidopsis seedlings, respectively, using the primers described in Supplementary Table 1 (AtD14-F-blunt and AtD14-R EcoRI for AtD14, OsD14-F-blunt and OsD14-R EcoRI for OsD14). For OsD14, the conserved esterase domain lacking the N-terminal region (residues 1–54) was used. The PCR products were digested by EcoRI and cloned into SmaI and EcoRI sites of the modified pMALc5x (New England Biology) vector, containing a polyhistidine tag and HRV 3C protease site from pET49b (Novagen), to yield OsD14-pMALHis and AtD14-pMALHis. *E.coli* Rosseta-gami 2 (Novagen) was used for recombinant protein expression. Overnight cultures (10 mL) were inoculated to fresh LB medium (1 L) containing 50 µg/mL ampicillin, 50 µg/mL streptomycin, 12.5 µg/mL tetracyclin, and 34 µg/mL chloramphenicol. After $OD_{600}$ reached 0.6–1.0, 0.1 mM IPTG was added and the cells were further incubated at 16 °C for 20 h. The cells were pelleted by centrifugation, then resuspended and sonicated in a lysis buffer (20 mM Tris-HCl (pH 8.0), containing 150 mM NaCl). The supernatants from the resulting lysates were subjected to MBPTrap column chromatography (5 mL, GE Healthcare). After washing with the lysis buffer, bound proteins were eluted using elution buffer (Lysis buffer containing 10 mM Maltose). To obtain MBP-OsD14 or MBP-AtD14, the eluate was further purified by TALON column chromatography (GE Healthcare). After applying a wash buffer (20 mM Tris (pH 8.0) containing 300 mM NaCl and 2 mM Imidazole), bound proteins were eluted using an elution buffer (20 mM Tris (pH 8.0) containing 300 mM NaCl and 50 mM Imidazole). The eluate was concentrated using an Amicon Ultra-4 10 K (Millipore) and the concentration of purified protein was adjusted to 5 µg/µL. The recombinant protein solution was divided into aliquots, immediately frozen in liquid nitrogen, and stored at −80 °C until use. To prepare the untagged AtD14 protein, the eluate from the MBP Trap column (MBP-AtD14), was treated with HRV 3 C protease at 4 °C for 3 h. The resulting sample was diluted with 20 mM Tris (pH 8.0) buffer and subjected to Hi trap Q HP anion exchange column chromatography (GE Healthcare). Absorbed proteins were eluted across a linear gradient of 0–700 mM NaCl in 20 mM Tris (pH 8.0). The fractions containing AtD14 were collected and concentrated using an Amicon Ultra-4 10 K (Millipore). The concentration of recombinant protein was adjusted to be 5 µg/µL and proteins were divided into aliquots, immediately frozen in liquid nitrogen, and stored at −80 °C until use. For AtD14, we used untagged protein in our initial experiments (Supplementary Fig. 1 and 2b-d), although we later used MBP fusion proteins (Supplementary Fig. 2e, and all the DSF experiments including time-course analysis) because no significant differences were found in the kinetics of the untagged versus MBP fusion proteins (Supplementary Table 2). For OsD14, we only used the MBP fusion protein due to its low solubility after removal of the MBP tag. Each point mutation construct was prepared using KOD-Plus-Mutagenesis Kit (TOYOBO). The nucleotide sequences for the mutant constructs are described in Supplementary Table 1. Protein expression and purification for the mutants was carried out as described above.

**Hydrolase activity tests of D14.** Hydrolase activity tests of AtD14 and OsD14 were carried out at 30 °C for 15 min in 100 µL of a standard reaction buffer that containing 10 µg of recombinant protein, 1 µM (or 10 µM) of substrates in 50 mM Phosphate-Na buffer (pH 7.0) containing 2% acetone. The enzyme reaction was stopped by the addition of 100 µL of acetonitrile, and ABC-FTL, HMB, and remaining substrate was analyzed by LC-MS/MS. For the analysis of hydrolase activity with 5DS, GR24, orobanchol, and GR7 each corresponding formyllactone part was analyzed by using each deuterium labeled standard as an internal standard, and the exact amount of reaction product was calculated. For the analysis of hydrolase activity with dhGR24, GR5, Br-PMF, and CN-PMF, a common reaction product, HMB was analyzed by LC-MS/MS and its peak area was used for the calculation of relative activities. The kinetic parameters were calculated from Lineweaver-Burk plots. The effect of substrate concentration on reaction velocity was examined at various concentration of 5DS (1.25, 2.5, 5, 10, 20 µM). Detailed conditions for the LC-MS/MS analysis are described in Supplementary Table 3.

**Differential scanning fluorimetry experiments.** DSF experiments were carried out using Mx3000P (Agilent). Sypro Orange (Ex/Em: 490/610 nm, Invitrogen) was used as the reporter dye. Reaction mixtures were prepared in 96-well plates, and each reaction was carried out on a 20 µL scale in PBS buffer containing 10 µg protein, SLs in acetone so that the final acetone concentration was 5%, and 0.015 µL Sypro Orange. In the control reaction acetone was added instead of the chemical solution. Samples were heated from 25 °C to 95 °C after incubation at 25 °C for 10 min in the absence of light. The denaturation curve was obtained using MxPro software.

**Time-course DSF experiments and hydrolysis monitoring.** MBP-AtD14 (24 µg) was incubated in 15 µL of PBS (pH7.4), and at each time point (0, 120, 180, 210, 225, and 240 min, for GR24. 0, 120, 240, 300, 330, 360 min for CN-PMF), 30 µL of GR24, or CN-PMF, in PBS (7.5% acetone) was added to the protein solution to initiate the reaction. The final concentration of the chemicals was 40 or 200 µM for

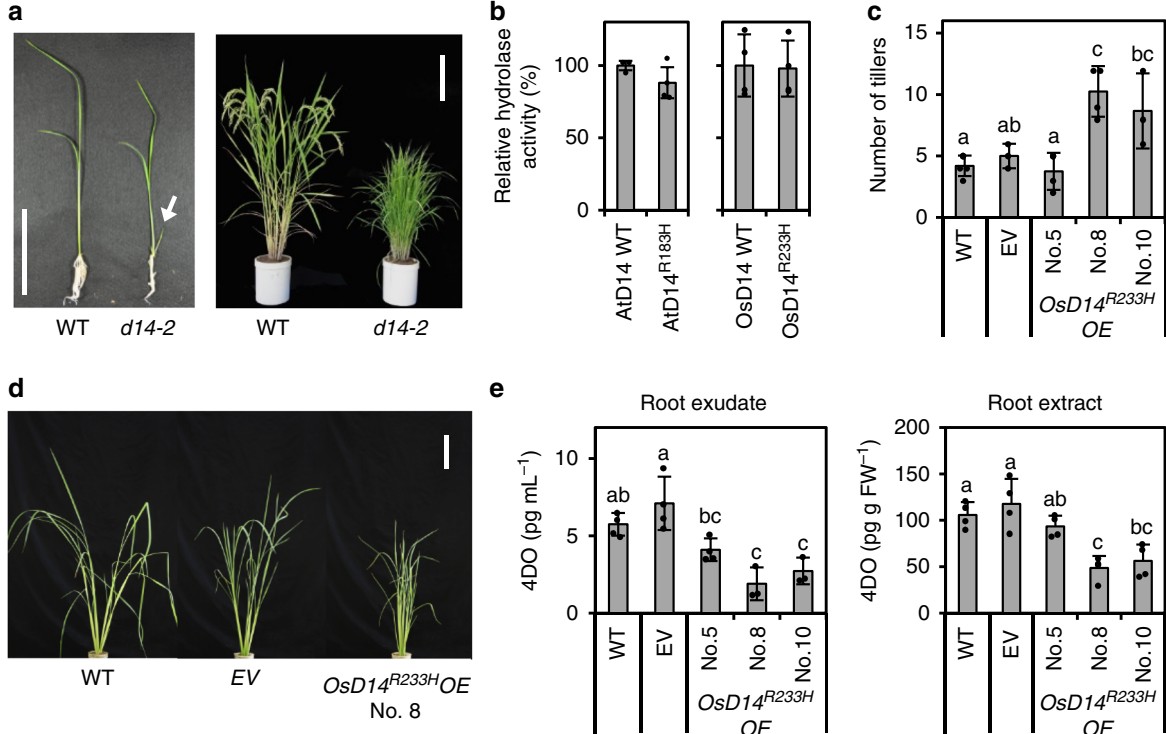

**Fig. 4** In vitro and in vivo functional analysis of OsD14$^{R233H}$/AtD14$^{R133H}$. **a** Phenotypes of 2 weeks old seedlings (left) and mature plants (right) of the rice *d14-2* mutant. The white arrow in the left picture indicates the outgrowing tiller. Scale bars = 5 cm (left panel), 20 cm (right panel). **b** Hydrolase activities of OsD14$^{R233H}$ and AtD14$^{R133H}$ mutants using 1 μM 5DS as a substrate. Data are the means ± SD (n = 3–4). **c** No. of tillers of rice transgenic plants overexpressing OsD14$^{R233H}$ in the WT (Nipponbare) background. Empty vector expressing plants are indicated as EV. Data are the means ± SD (n = 3–5). **d** Phenotypes of 42 days old transgenic plants overexpressing OsD14$^{R233H}$ (*OsD14$^{R233H}$OE*). Scale bars = 10 cm. **e** Quantitative analysis of 4DO in the root exudates (left panel) and extracts (right panel) of OsD14$^{R233H}$ overexpressing (*OsD14$^{R233H}$OE*) plants. Data are the means ± SD (n = 3–4). Different letters in **c** and **e** indicate significant differences at *P* < 0.05 with Tukey-kramer multiple comparison test. Source data are provided as a Source Data file

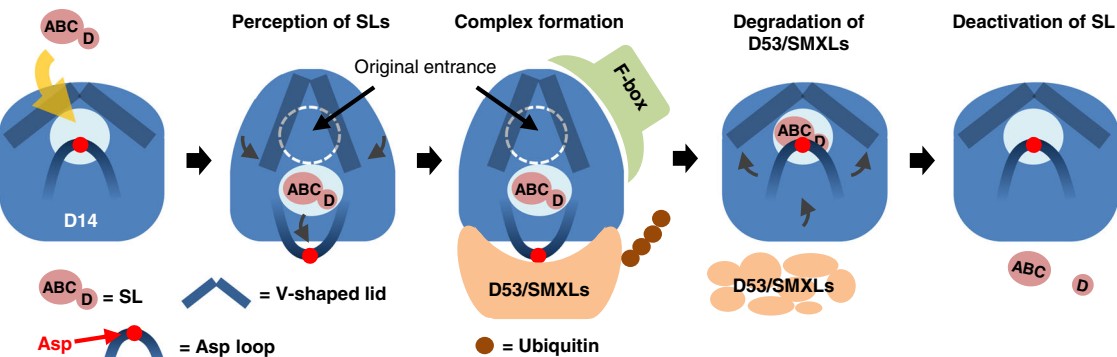

**Fig. 5** A proposed working model of D14 in the SL signaling pathway. A bioactive SL molecule induces the protein conformational changes of D14, which triggers complex formation with the signaling partners. After the degradation of D53/SMXLs and transmission of the SL signal, D14 reconstructs the catalytic triad to hydrolytically decompose the bioactive SL

both chemicals. After 4 h (for GR24) or 6 h (for CN-PMF) incubation at 30 °C, the reaction was terminated and 15 μL and 25 μL of each sample were used for DSF experiments and hydrolysis analysis, respectively. For DSF experiments, 5 μL of Sypro Orange solution, diluted 333X with PBS, was added to each sample. DSF experiments were performed as described above. For hydrolysis assays, the reaction was stopped by adding 25 μL of acetonitrile. For GR24 type ABC-FTL, $d_1$-GR24 type ABC-FTL was used as an internal standard. For HMB, HBN, and CN-PMF, 1-naphtaleneacetic acid (NAA) was used as an internal standard. The detailed analysis condition for each chemical is described in Supplementary Table 3.

**Analysis of the AtD14 protein modification during hydrolysis**. MBP-AtD14 (24 μg) was incubated in 45 μL of PBS (pH 7.4) containing a chemical (GR24 (40 μM), CN-PMF (40 μM or 200 μM), or HMB (40 μM or 400 μM)) and 5% acetone. For the native condition analysis, the reaction mixture was directly applied to LC-MS/MS

analysis. As for the denatured condition, acetonitrile was added at 50% final concentration to each sample to terminate the reaction. LC-MS/MS analysis of MBP-AtD14 was carried out using a system consisting of a quadruple/time-of-flight tandem mass spectrometer (TripleTOF 5600, AB SCIEX) and an Ultra high performance liquid chromatograph (Nexera, Shimadzu). The detailed analysis condition is described in Supplementary Table 3. BioAnalyst software was used for acquisition and data processing including deconvolution of multiply charged ions.

**Generation of transgenic plants**. The cDNA of *OsD14* and *AtD14* was obtained by PCR amplification using primers as described in Supplementary Table 1 (AtD14-F-cacc and AtD14 R-blunt for AtD14, OsD14-F-cacc and OsD14-R-blunt for OsD14), and each PCR product was subcloned into the entry vector pENTR/D-TOPO (Invitrogen). For *OsD14*, the conserved esterase domain was used, as described above. Each point mutation construct was generated by PCR using

mutant constructs, which were used for protein expression as described above, as templates, and subcloned into the same entry vector. Each cDNA was shuttled into the 35S promoter vector, pGWB2[32], by an LR clonase reaction according to the manufacture's protocols (Invitrogen). Arabidopsis WT, atd14-2, and atd14-2 max4-8 plants were transformed with the resulting constructs by the floral dip method using Agrobacterium tumefaciens. The rice Nipponbare or osd14-1N mutant plants were transformed with the resulting constructs by A. tumefaciens. For genetic complementation of the osd14-2 mutant using a native promoter, a 4,507-bp genomic fragment containing the OsD14 (Os03g0203200) gene, as well as regions 2 kb upstream and 1 kb downstream of the transcribed region, was amplified by PCR using primers described in Supplementary Table 1 (OsD14-genome-F and OsD14-genome-R). The PCR product was subcloned into the entry vector pENTR/D-TOPO. The fragment was then shuttled into a binary vector containing no promoter, pGWB1[32], by an LR clonase reaction according to the manufacture's protocols. Rice osd14-2 mutant were transformed with resulting constructs by A. tumefaciens.

**Arabidopsis shoot branching assay**. Arabidopsis seeds were sterilized in a 1% sodium hypochlorite solution for 5 min, rinsed in sterile water, and stratified for one day at 4 °C. The seeds were placed on half strength Murashige and Skoog (MS) medium containing 1% sucrose and 1% agar (pH 5.7) at 22 °C under fluorescence white light (60–70 µMol m$^{-2}$s$^{-1}$) with a 16 h light/8 h dark photoperiod for 15 days. Plants were then transferred to a glass pot containing 400 mL hydroponic solution and grown under the same environmental conditions for an additional 15 days. The solution was renewed every 7 days.

**Quantitative analysis of endogenous SLs in rice plants**. In order to quantitatively analyze the endogenous SLs, we used a hydroponic culture system for growing rice. Rice seeds were washed in 70% ethanol for 30 s, sterilized in 2.5% sodium hypochlorite solution for 15 min, rinsed with sterile water, and then incubated in sterile water at 28 °C under fluorescence white light (150 µMol m$^{-2}$s$^{-1}$) with a 16 h light/8 h dark photoperiod for 5 days. Each seedling was then transferred to a glass vial containing a sterilized hydroponic culture solution (13 mL, −P), and grown under the same condition for an additional 7 days (total 2 weeks). For transgenic plant selection (T1 generation), genotyping was carried out 2 days after transfer using primers described in Supplementary Table 1 (pGWB2-F and pGWB2-R), so that only the transgenic plants were picked. The plants were allowed to continue growing and were subsequently used for quantitative analysis. Two weeks after germination, the hydroponic culture media were collected, the internal standard was added, and the media was extracted twice with ethyl acetate. The ethyl acetate phase was evaporated to dryness under nitrogen gas and redissolved in ethyl acetate:n-hexane (15:85). The extracts were loaded onto Sep-pak Silica 1 mL cartridges (Waters), washed with ethyl acetate:n-hexane (15:85) and then eluted with ethyl acetate:hexane (50:50). The eluates were evaporated to dryness and dissolved in 50% acetonitrile, then subjected to LC-MS/MS analysis. To analyze 4DO in root samples, the roots (0.5–1.0 g) were homogenized in 10 mL of acetone after addition of the internal standard. The filtrates were evaporated to dryness under nitrogen gas, redissolved in deionized water, and extracted with ethyl acetate twice. The ethyl acetate phase was evaporated to dryness under nitrogen gas. The extracts were then dissolved in 20% acetone and loaded onto Oasis HLB 1 mL cartridges (Waters) and eluted with 50% acetone after washing with 20% acetone. The eluates were evaporated to dryness under nitrogen gas, dissolved in ethylacetate:n-hexane (15:85), and loaded onto Sep-pak Silica 1 mL cartridges (Waters). The column was washed with ethyl acetate:n-hexane (15:85) and then eluted with ethyl acetate:n-hexane (50:50). The purified 4DO-containing fractions were evaporated to dryness then dissolved in 50% acetonitrile and subjected to LC-MS/MS analysis.

LC-MS/MS analysis of 4DO was carried out using a system consisting of a quadruple/time-of-flight tandem mass spectrometer (TripleTOF 5600, AB SCIEX) and an Ultra high performance liquid chromatograph (Nexera, Shimadzu) equipped with a reverse phase column (Acquity UPLC BEH-C18, φ2.1 × 50 mm, 1.7 µM; Waters). Detailed analysis conditions are described in Supplementary Table 3.

**Western blot analysis of AtD14 protein in Arabidopsis**. Arabidopsis plants were grown on half strength MS agar plates for 15 days as described above. The aerial part of plants were harvested and immediately frozen by liquid nitrogen, then ground using a motor and pestle, and resuspended in 50 mM Tris buffer (pH 7.5) containing 10 mM MgCl$_2$, 150 mM NaCl, 0.1% Triton, 1% protease inhibitor cocktail (Sigma), 1 mM PMSF, and 1 mM DTT. The tissue slurry was removed by centrifugation, and the supernatant was used for the analysis after the concentration was adjusted. Protein samples (each 75 µg) were separated by 15% SDS-PAGE and transferred to nitrocellulose membrane. For the detection of AtD14 protein, the blots were treated with 5% skim milk in TBST (0.1% Tween20 in 2 mM Tris-HCl pH 7.6, and 13.7 mM NaCl) for 2 h and subsequently incubated with anti AtD14 antibody at 25 °C overnight. AtD14 antibody was produced in rabbit by BioGate Co using recombinant AtD14 protein (GST fusion) as the antigen. The quality of the antibody was evaluated using recombinant AtD14 and KAI2 (Supplementary Fig. 12). After washing three times with TBST (10 min each), the membrane was incubated with goat anti-rabbit IgG horseradish peroxidase-conjugated secondary antibody (Sigma, A4914, 100 times dilution) at 25 °C for 2 h.

The membrane was washed three times and analyzed by a chemiluminescence-based detection method using Super Signal West Pico Chemiluminescense Substrate (Pierce).

**Yeast two (three) hybrid experiments**. The WT and each mutant AtD14 were cloned into pGADT7, and SMXL7 was cloned into pGBKT7[22]. Arabidopsis ASK1 was amplified by PCR using primers, ASK1-F-cacc and ASK1-R-blunt (Supplementary Table 1), and the PCR product was subcloned into the entry vector pENTR/D-TOPO. ASK1 was then shuttled into the modified pYES-DEST52 (pYES ADHPro)[33] vector by an LR clonase reaction according to the manufacture's protocols (Invitrogen). We were not able to clone Arabidopsis MAX2 into pGBKT7, thus MAX2 was cloned into a modified pGBDU, in which the URA3 marker region was replaced with TRP1 marker. To modify pGBDU, the TRP1 region including its promoter was amplified with primers, TRP1-Pro-F-NdeI and TPR1-R-NcoI (Supplementary Table 1), using pGBKT7 as a template. The PCR product was digested by NdeI and NcoI, and cloned into the corresponding enzyme sites of pGBDU. We named the resulted vector pGBDT. MAX2 was amplified by PCR using primers, MAX2-F-BamHI and MAX2-R-PstI (Supplementary Table 1), and the PCR product was cloned into the corresponding enzyme sites of pGBDT. Resulting constructs were co-transformed with the yeast strain PJ69-4A and the transformants were grown on SD-Trp/-Leu or SD-Trp/-Leu/-Ura plates for 3 days at 30 °C. Interactions between the two proteins were examined on selective media (SD-Trp/-Leu/-His or SD-Trp/-Leu/-Ura/-His) containing a 1:1000 dilution of the tested SL from an acetone-dissolved stock solution (0.1% acetone was used as a control). The plates were kept for 5 days at 30 °C. For western blotting analysis, each transformant was pre-incubated in SD-Trp/-Leu overnight at 30 °C, transferred to fresh YPDA medium, and incubated until the OD$_{600}$ reached 0.5–0.6. The cells were collected by centrifugation then resuspended with an Urea/SDS protein extraction buffer (40 mM Tris (pH 6.8) containing 8 M Urea, 5% SDS, 0.1 mM EDTA, 0.4 mg/mL Bromophenol blue, and 1 mM PMSF). Each sample was transferred to a new tube containing zirconia beads, and the samples were vortexed vigorously for 1 min. After centrifugation at 14,000 rpm for 5 min at 4 °C, the supernatants were transferred to new tubes. After the protein concentration was adjusted using a Protein Quantification Assay kit (Takara), protein samples (each 1 µg) were separated by 10% SDS-PAGE and transferred to a nitrocellulose membrane. For the detection of AtD14 protein fused with the GAL4 activation domain, the blots were treated with 5% skim milk in TBST (0.1% Tween20 in 2 mM Tris-HCl, pH7.6, and 13.7 mM NaCl) for 2 h and subsequently incubated with GAL4 AD Monoclonal Antibody (Clontech, 630402, 4000 times dilution) at 25 °C overnight. After washing three times with TBST for 10 min each wash, the membrane was incubated with goat anti-mouse IgG horseradish peroxidase-conjugated secondary antibody (abcam, ab97023, 2000 times dilution) at 25 °C for 2 h. The membrane was washed three times, and analyzed by a chemiluminescence-based detection method using Super Signal West Pico Chemiluminescense Substrate (Pierce).

**SL-binding assay**. The direct binding assay of AtD14 with 5DS was performed according to the GA-binding assay method[34]. MBP-AtD14 was incubated at 30 °C for 30 min in 100 µL of a standard binding buffer that containing 50 µg of recombinant protein and 20 µM of 5DS in 50 mM Phosphate-Na buffer (pH 7.0) containing 150 mM NaCl and 2% acetone. After 30 min incubation, the sample was loaded onto the NAP-5 column chromatography (GE Healthcare). The column was eluted by 100 mM Phosphate-Na buffer (pH 7.0) containing 150 mM NaCl, and the first 800 µl eluents were collected. d$_1$-rac-5DS was added to each sample as an internal standard. The samples were extracted by 200 µL of Ethyl acetate, and the Ethyl acetate layer was evaporated using the nitrogen gas. Each sample was dissolved with 50% of acetonitrile/water, and analyzed using LC-MS/MS.

**Reanalyzing of the structurally changed AtD14**. The PDB data of AtD14-D3-ASK1 complex structure (5HZG) were downloaded from protein data bank (https://www.rcsb.org/), and reanalyzed by using PyMOL. The cavity volume was calculated using CASTp program server (http://sts.bioe.uic.edu/castp/index.php). Docking was performed using SWISS DOCK (http://www.swissdock.ch//).

**qRT-PCR analysis of rice transgenic plants**. Rice plants were grown in a growth chamber at 25 °C with a 16 h light/8 h dark photoperiod for 6 weeks. Genotyping was carried out to select the transgenic plants, using primers described in Supplementary Table 1 (pGWB2-F and pGWB2-R). Total RNA was extracted from leaf blades of the plants using a Plant RNA Isolation mini kit (Agilent). After DNase I treatment, first-strand cDNA was synthesized using SuperScript III reverse transcriptase (Invitrogen). The primer sets used to amplify the transcripts were described in Supplementary Table1 (OsD14-QRT-F and OsD14-QRT-R). PCRs were performed with SYBR green I using a Light Cycler ® 480 System II (Roche Applied Science).

**Reporting summary**. Further information on experimental design is available in the Nature Research Reporting Summary linked to this article.

## Data availability

The source data underlying Fig. 2b, d, 3a,b,e, and 4b,c,e and Supplementary Figs 1c, 2b-e, 3b-e,g,i, 4a-e, 5a, 6a-c, 7c, 8b-d, and 11b-g are provided by a Source Data file. A reporting summary for this Article is available as a Supplementary Information File. All other data are available from the corresponding authors upon a reasonable request.

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

## Acknowledgements

This work was supported by JSPS/MEXT KAKENHI (Grant Nos, 24780117, 24114010, and 17H06474), the Program for Promotion of Basic and Applied Research for Innovations in Bio-Oriented Industry, and the Core Research for Evolutional Science and Technology (CREST) by JST (Grant No, JPMJCR13B1). Y.S. was a JSPS Postdoctoral Fellow for Research Abroad. We thank Dr. Mitsunori Seo for providing the yeast expression vector, pYES ADHpro.

## Author contributions

Y.S. and S.Y. designed the research. Y.S. and R.Y. performed the majority of experiments with the guidance from K.M. and S.Y. H.K. generated the rice transgenic plants and analyzed the phenotypes of these plants with guidance from J.K. C.M. performed part of western blot analysis. C.M. and E.S. performed part of Y2H experiments. R.H. performed Y3H experiments. A.S. performed part of Arabidopsis transgenic plants preparation and part of hydrolysis assays. M.T. performed the characterization of rice osd14-2 mutant with the guidance from R.T. M.U. technically supported rice and Arabidopsis plant growth. A.H. and T.K. technically supported the LC-MS/MS analysis. K.A. prepared SL chemicals. N.T-K. and W.L. generated Arabidopsis atd14-2 max4-8 double mutant. Y.H, and T.H. supported the protein expression work. Y.S. and J.B. analyzed the reported structural data with guidance from J.P.N., Y.S., R.Y., J.B., and S.Y. wrote the manuscript.

## Additional information

**Competing interests:** The authors declare no competing interests.

