## [Peer Review File · Nature Communications]

Reviewers' comments:

Reviewer #1 (Remarks to the Author):

The paper entitled Strigolactone perception and deactivation by a hydrolase receptor, DWARF14 by Seto, Yamaguchi et al. deals with the elucidation of the perception mechanism of Strigolactones, the recently discovered class of plant hormones. The issue is a matter of debate among scientists and so far, the uncertainty about the mechanism of perception still seriously hampers the full development of SLs potential in agricultural applications.

The paper is timing as, a recently paper by Carlsson et al. instills doubt about the X-RAY structures of the receptor with the putative ligand trapped inside, present in the literature. Many research groups, very active in the field grounded their research and experiments on the hypothesis that a part of the SL, the D-OH ring (HMB) remains trapped inside the receptor. This might be completely misleading. Eventually, the authors with this paper shed light on this topic and demonstrated that the event responsible for signaling transmission is an early event and it is independent from hydrolysis of the substrate, furthermore the two events (perception and hydrolysis) are genetically and biochemically separated.

The experiments are soundly and strongly support the hypothesis put forward

The manuscript is well written, concise and rational, straightforward and clear.

I attached the PDF file of the manuscript with my comments and suggestions.

Reviewer #2 (Remarks to the Author):

SL are phytohormones that are sensed by the receptor D14, an α/β -hydrolase. Upon sensing, D14 directly interacts with D3SCF and a family of transcriptional repressors that include SMXL7. This brings D3SCF and SMXL7 into proximity to allow repressor ubiquitination and proteolysis to relieve expression of SL target genes. Since D14 can also slowly hydrolyze SL, the mechanisms of reception and signal transduction have been an area of considerable confusion. A recent crystal structure of the complex between D14 and its effector D3 with a covalently bound SL hydrolysis intermediate (CLIM) suggested a novel mechanism, in which CLIM would induce closure of the D14 α -helical ligand

binding gate, which in turn would generate a binding site for the D3 F-box subunit of D3SCF. In the submitted manuscript, Seto et al. challenge this model, most importantly by demonstrating that a catalytically inactive D14 mutant can complement a d14 mutant and can interact with SMXL7 in a Y2H assay. They propose a model in which the intact SL induces conformational changes in D14 that mediate D3 interaction, while the subsequent SL hydrolysis would lead to signal termination. The results are interesting to a broad readership, but major concerns need to be addressed before I can recommend this paper for publication.

Major Concerns:

1. The authors discredit the existence of a relatively long-lived, covalently bound SL intermediate (CLIM). They base this on i) an alternative interpretation of the CLIM density in the crystal structure proposed by another group, and ii) on their time course experiment, which demonstrates a close match between the time of SL hydrolysis and the reversal of a SL-induced thermal D14 destabilization. They take this reversal as evidence for immediate SL product release, an interpretation that is likely wrong. While the CLIM density in the crystal structure alone is not sufficient as CLIM evidence, two independent groups have provided strong MS evidence for the existence of CLIM, which has been further supported by biochemical data. Together, these data provide much more compelling evidence for CLIM than the authors' indirect counter evidence. If the authors want to make a claim for immediate hydrolysis release, they need to provide direct evidence, for instance by MS.
2. The crystal structure clearly shows formation of the D14-D3 signaling complex in the absence of an intact SL, which is difficult to reconcile with the authors' model.
3. A hallmark of SL perception is the SL-induced thermal destabilization of D14, which has been hypothesized to be a consequence of a D14 conformational rearrangement required for D3 binding. The authors provide important and convincing evidence that thermal destabilization is induced by intact SL, and reversed upon SL hydrolysis (note that there is an important discrepancy on how the experiment is performed between text/figure legend and the Methods section). However, thermal re-stabilization does not exclude CLIM formation (see above), nor is there evidence that thermal destabilization coincides with D3 binding.
4. Endogenous and synthetic SL can have very different chemical scaffolds. Extensive structure-function analysis has identified as minimum requirement the SL D-ring moiety attached by a labile, hydrolyzable bond to a variety of scaffolds. Also, limited previous attempts to mutationally separate D14 SL-responsiveness from SL hydrolysis had been unsuccessful, in part because residues important for SL catalysis have also been important for SL binding. Here the authors provide evidence that an alanine mutation of the catalytic triad residue D218 (which does not directly bind SL) severely compromises SL hydrolysis, as expected, yet as an overexpressed protein is still capable to

complement a d14 mutation and to interact with SMXL7 in a Y2H assay. SMXL7 and D3 can both independently bind to D14 at supposedly two different surfaces, but can also directly bind each other. To gain mechanistic insight, it would therefore be important to see whether D14 D218A retains the ability to interact with D3 in a SL-dependent manner (Y2H or other assay).

5. The authors identified a new d14 mutant, R233H (d14-2), located in the helical lid domain. D14 R233H is catalytically active, but failed to complement d14 and has a weakened Y2H interaction with SMXL7. Overexpression of D14 R233H in wildtype cells resulted in increased branching, suggesting that it retains the ability to hydrolyze endogenous SL without inducing SL signaling. The authors interpret this phenotype as support for their hypothesis that SL hydrolysis is a deactivating step. Mutation of the same residue (R183E in their aa labelling) has previously been demonstrated to disrupt D3 binding (Zhao et al, 2015), while several groups have shown that the SL D-ring alone has very low D14 binding affinity and would not bind D14 under physiological conditions. The mutational phenotype can therefore simply be explained by the inability of the D14 R233 mutant to interact with D3 (and therefore to signal), while it remains active in hydrolyzing endogenous SL which would prevent their ability to bind D14 and form covalently bound reaction intermediates (CLIM).

In summary, the manuscript provides unexpected and important new insight indicating that the current model of SL signaling is incomplete, yet falls short in providing the evidence for the proposed alternative model.

Reviewer #3 (Remarks to the Author):

This manuscript described biochemical and genetic experiments on the detailed mechanism of strigolactone (SL) signalling by the D14 receptor. The authors argue that their work overturns the current model of SL signalling, and that SL signalling does not require hydrolysis of SL molecules.

My overall impression is that this is very important data, which needs to be published. However, I feel that the preparation of the manuscript is little bit rushed and incoherent. There is a huge of data here (especially in supplementary figures), but I'm not sure it has been put together in a particularly logical way. Some experiments are not at all well described, and just appear 'out of the blue' within the manuscript.

I have a few major suggestions for improving the manuscript.

1) In the first section of the manuscript (e.g. Figure 2) strongly suggest that SL cleavage is not needed for signalling, I do not think the data are completely unambiguous, because the level of signalling still correlates with the ability of the protein to cleave SLs. So when there is no more GR24 to cleave, the signalling stops. In CN-PMF, the substrate is still cleaved, at a slower rate, and signalling is less. There it still substrate at 6 hours, and so signalling continues. The data here do not completely rule out cleavage being necessary for signalling. While I find the data are convincing, I think the authors need to be slightly careful about the strength of their interpretation from these experiments. For instance, line 231: "our data clearly demonstrate that the uncleaved substrate itself not the hydrolysis intermediate or the products, induces the Tm shift of AtD14". To me, the authors data do suggest this, but I disagree that they 'clearly' demonstrate this.

Furthermore, the authors say that they have an completely undegradable SL (3,6'-dihydroGR24), but they say that is biologically inactive. This suggests that cleavage IS necessary for signalling. If the authors' model is correct 3,6'-dihydroGR24 should still trigger signalling. So the authors should need to show whether 3,6'-dihydroGR24 triggers signalling by DSF. The only other explanation is that 3,6'-dihydroGR24 does not bind D14 at all - in which case it is not surprising it is neither biologically inactive nor cleaved by D14 - but which would make it completely irrelevant to the paper. Either way, this issue needs resolving.

2) In the second section of the manuscript, the authors present evidence that D14 is NOT covalently modified by GR24, and remains available to act up on more GR24. This is based on D14 being able to metabolize all GR24 despite there being a large molar excess of GR24. I feel that the experiment presented in Fig 2B could be made more convincing if the authors 'spike in' fresh GR24 after 4 hours, and show that the original D14 protein is still able to act up on this fresh ligand/substrate.

3) In the third section of the manuscript, I found it quite strange that some catalytic triad mutants could still partially rescue d14, but not all of them. Clearly, the ability of the D218A mutant to rescue d14 supports the authors' model that cleavage is not necessary for signalling. I don't dispute the data, but the authors do not provide a convincing explanation for this observation for the differences between the catalytic mutants, which I find weakens their argument.

There is only a statement that "Therefore, the mutation to Ser and His possibly affected the initial interaction with SLs." I would find the manuscript more convincing if the authors could provide some evidence data (perhaps from modelling protein structure) to show that this explanation (i.e. that some catalytic triad mutants don't bind SLs, let alone cleave them) is correct.

Again, I disagree with the statement at line 290:

"These results, together with the time-course DSF experiments, conclusively demonstrate that the hydrolase reaction catalyzed by D14 is not necessary for the signal transducing role.". I don't think the data is completely conclusive, because there are still catalytic mutants that cannot perform signalling, for which the authors have not provided a convincing explanation.

4) In the fourth section of the manuscript, the structural biology work appears completely out of the blue without any explanation. The authors simply say:

"When we reanalyzed the structural changes in AtD14".

There are no details in the M&M as to how the authors did this. Nor do they provide any in the results section. It is completely unclear whether this is an in-silico reassessment of structure, or if they have recrystallized the protein. How did the authors do this? Why did they do it? Why do they find different results to previously published. We need to know all these things.

Also, I think these data are primary data and should be in a main figure, because they would represent a major change in our understanding of D14 structure and function.

5) In the fifth section, the last part of the section is very poorly explained. I didn't follow the logic of this at all.

"We also found that the levels of an endogenous SL, 4-deoxyorobanchol (4DO), in the rice overexpressors were decreased relative to WT and the empty vector-expressing plants (Fig. 4e). If considering the quite weak or no biological activities with SL hydrolysis products our data suggest that the hydrolase reaction catalyzed by D14 would be a deactivating step of SLs after transducing the signal."

This needs rewriting or removing.

Minor points:

1) Line 105: I don't think we really do have a 'near complete picture' of SL synthesis. There is still lots unknown.

2) Line 116: D53/SMXL7 is not 'a signalling suppressor' protein. The signalling pathways ends with degradation of D53, there is nothing for D53 to suppress. It might be a 'transcriptional repressor', but that is not the same as a 'signalling suppressor'.

3) Line 136: Gunilla et al is claimed as reference 21, but that is not what is stated in the reference list.

4) Line 171: Br-PMF and CN-PMF are not introduced properly. Please state what they are.

5) Line 182: GR5 is not introduced at all. Please state what it is.

6) "the second attack from a water molecule must be occurred immediately after the first attack from"

This sentence appears out of nowhere, without any previous discussion of the chemical mechanism of D14 catalytic activity. It also makes no sense unless the reader has a background in chemistry. Please modify.

7) Line 219 "denoting"  suggesting

8) Line 275 "a little far from it". This is not a scientific statement. Define this better.

9) Line 317: Given the manuscript already uses Atd14-2, it is quite confusing that the authors name their new rice mutant d14-2. To avoid confusion, can they please refer to this new mutant as Osd14-2, to make it completely clear what is being referred to.

10) Line 336. Can the authors perhaps summarise here the evidence that suggests D14 degrades SLs after signal transmission?

11) Line 339: Why is this model in a supplementary figure, it is clearly the main finding of the paper? Replace Figure 5 with this figure.

12) Line 355: Where is the evidence to this statement? The data presented in supplementary figure 10 only provide very circumstantial evidence to support this idea.

13) In my opinion, the authors should not be continuing to introduce data during the discussion. Either supplementary figure 10 is important enough to mention during the results section, or it is not that important. I don't think supplementary figure 10 helps the authors arguments, and should be cut.

14) Figure 3e. In the legend, the panels are described as 'upper' and 'lower', but they are next to each other.

Response to the reviewer's comments

The corrections according to the comments from Reviewer #1 are reflected in blue color, and all the other corrections are reflected in red color in the revised manuscript.

Reviewer #1 (Remarks to the Author):

The paper entitled Strigolactone perception and deactivation by a hydrolase receptor, DWARF14 by Seto, Yamaguchi et al. deals with the elucidation of the perception mechanism of Strigolactones, the recently discovered class of plant hormones. The issue is a matter of debate among scientists and so far, the uncertainty about the mechanism of perception still seriously hampers the full development of SLs potential in agricultural applications.

The paper is timing as, a recently paper by Carlsson et al. instills doubt about the X-RAY structures of the receptor with the putative ligand trapped inside, present in the literature. Many research groups, very active in the field grounded their research and experiments on the hypothesis that a part of the SL, the D-OH ring (HMB) remains trapped inside the receptor. This might be completely misleading. Eventually, the authors with this paper shed light on this topic and demonstrated that the event responsible for signaling transmission is an early event and it is independent from hydrolysis of the substrate, furthermore the two events (perception and hydrolysis) are genetically and biochemically separated.

The experiments are soundly and strongly support the hypothesis put forward. The manuscript is well written, concise and rational, straightforward and clear. I attached the PDF file of the manuscript with my comments and suggestions.

→Response

Thank you so much for favorable comments on our manuscript. We have revised our manuscript according to the comments on the pdf file, which are reflected in blue in our revised manuscript. Please find below an answer for a specific comment.

As for the naming of the new *d14* mutant allele.

In the original paper of Arabidopsis D14 (Waters et al, Development, 2012), the authors named it *Arabidopsis thaliana D14 (AtD14)* because it was discovered by a reverse genetic approach based on the sequence homology with the initially characterized rice D14 (Arite et al. Plant Cell Physiology, 2009). The mutant was named *Atd14* in this paper. According to this paper, we are using *atd14* for the Arabidopsis *d14* mutant naming (we are not sure which is proper, *Atd14* or *atd14*). In the original paper, the authors characterized the first allele, *Atd14-1*, and we characterized the second allele, *atd14-2* (Seto et al, PNAS, 2014), which is used in this study. Thus *d14-#* should be correct for the rice mutant allele naming, and *atd14-#* should be for the Arabidopsis

mutant. In our manuscript, a new allele was characterized from rice, thus we named it *d14-2*, which is different from *atd14-2*. However, we also agree that this naming is very confusing, thus we have decided to use *osd14* for the rice mutant naming in our manuscript to distinguish from *atd14*.

Reviewer #2 (Remarks to the Author):

SL are phytohormones that are sensed by the receptor D14, an α/β -hydrolase. Upon sensing, D14 directly interacts with D3SCF and a family of transcriptional repressors that include SMXL7. This brings D3SCF and SMXL7 into proximity to allow repressor ubiquitination and proteolysis to relieve expression of SL target genes. Since D14 can also slowly hydrolyze SL, the mechanisms of reception and signal transduction have been an area of considerable confusion. A recent crystal structure of the complex between D14 and its effector D3 with a covalently bound SL hydrolysis intermediate (CLIM) suggested a novel mechanism, in which CLIM would induce closure of the D14 α -helical ligand binding gate, which in turn would generate a binding site for the D3 F-box subunit of D3SCF. In the submitted manuscript, Seto et al. challenge this model, most importantly by demonstrating that a catalytically inactive D14 mutant can complement a *d14* mutant and can interact with SMXL7 in a Y2H assay. They propose a model in which the intact SL induces conformational changes in D14 that mediate D3 interaction, while the subsequent SL hydrolysis would lead to signal termination. The results are interesting to a broad readership, but major concerns need to be addressed before I can recommend this paper for publication.

Major Concerns:

1. The authors discredit the existence of a relatively long-lived, covalently bound SL intermediate (CLIM). They base this on i) an alternative interpretation of the CLIM density in the crystal structure proposed by another group, and ii) on their time course experiment, which demonstrates a close match between the time of SL hydrolysis and the reversal of a SL-induced thermal D14 destabilization. They take this reversal as evidence for immediate SL product release, an interpretation that is likely wrong. While the CLIM density in the crystal structure alone is not sufficient as CLIM evidence, two independent groups have provided strong MS evidence for the existence of CLIM, which has been further supported by biochemical data. Together, these data provide much more compelling evidence for CLIM than the authors' indirect counter evidence. If the authors want to make a claim for immediate hydrolysis release, they need to provide direct evidence, for instance by MS.

→Response

Thank you so much for a valuable and critical comment to improve our manuscript. We do not deny the occurrence of the covalently-linked intermediate during hydrolysis, but

the question is whether it is responsible for transducing the hormone signal. Our time-course analysis of the hydrolysis reaction showed that the first and second products released at almost the same speed, which strongly suggest that the covalently-linked intermediate has a very short life. On the other hand, there was a small difference in the release speed between ABC-CHO and HMB, and in consistent with this observation, we could also detect the covalently-linked intermediate of AtD14 by LC-MS/MS in the revised manuscript (Supplementary Fig. 4, line 262-290). We monitored this modified AtD14 peak over-time during hydrolysis, and found that the amount of the covalently-linked intermediate first increased from 0 to 15 min, which then gradually decreased as the hydrolysis reaction proceeded. After 4 h, when almost all the substrate was consumed, the modified AtD14 peak almost disappeared. Taken together, our data demonstrate that the D-ring part, which is once attached to AtD14, is promptly released as a second product. Moreover, these results show that the level of the covalently-linked reaction intermediate does not correlate with the T_m shift. Thus, we have concluded that the chemical substance inducing the melting temperature shift was the substrate itself.

2. The crystal structure clearly shows formation of the D14-D3 signaling complex in the absence of an intact SL, which is difficult to reconcile with the authors' model.

→Response

Thank you for pointing out that an intact SL was not present in the crystal structural data of the D14-D3-ASK1 complex. However, there is no evidence for the absence of intact SL in the crystal. Similarly, no evidence was provided for the existence of CLIM in the crystal because the proposed model of CLIM in the crystal was clearly shown to be crystallographically incorrect: the CLIM model does not fit to the electron density as was pointed out by Carlsson et al (*J. Exp. Bot.* 69, 2345-2354, 2018). In addition, it should be noted that identification of CLIM by MS/MS experiments was performed using a D14 protein sample prepared in solution but not a sample from the crystals. Thus, there was no direct evidence for the CLIM formation in the crystal.

Anyway, as the reviewer suggested, the clear electron density for the intact SL molecule was not detected in the reported structural data, most likely because of disordering of ligands. Indeed, the electron density for a loop region containing the catalytic triad Asp residue was not detected at all in this structure, possibly because of considerable flexibility of this region in the complex. Moreover, the crystallization was performed in a relatively high pH condition (Yao et al, *Nature*, 536, 469-473, 2016), which could induce a non-enzymatic hydrolytic degradation of the SL molecules in the pocket. Thus, we speculate that these factors could result in a diffused and poor electron density, which made it difficult to be interpreted with a single ligand model. Alternatively, it was also possible that an intact SL which induced the conformational change of D14 came off during crystallization or analysis procedure because of disordering of the Asp

loop region. We have added an explanation in our revised manuscript (line 434-440).

3. A hallmark of SL perception is the SL-induced thermal destabilization of D14, which has been hypothesized to be a consequence of a D14 conformational rearrangement required for D3 binding. The authors provide important and convincing evidence that thermal destabilization is induced by intact SL, and reversed upon SL hydrolysis (note that there is an important discrepancy on how the experiment is performed between text/figure legend and the Methods section). However, thermal re-stabilization does not exclude CLIM formation (see above), nor is there evidence that thermal destabilization coincides with D3 binding.

→Response

Thank you so much for a helpful comment. As we answered above, our LC-MS/MS analysis of the AtD14 protein enabled us to detect the covalently linked intermediate (Supplementary Fig, 4). However, a time-course analysis strongly suggested that such an intermediate does have a very short life. These results didn't exclude the presence of CLIM, but demonstrate that its lifespan is not as long as the proposed mechanism in the published papers (Yao et al, *Nature*, 536, 469-473, 2016; de Saint et al, *Nat. Chem. Biol.* 12, 787-794, 2016). As for the correlation between thermal destabilization and D3 binding, a previous report demonstrated that the biologically active SL analogs such as GR24 and debranones, but not inactive compounds such as ABC-FTL and HMB, could induce D14-D3 interaction as was examined by AlphaScreen assay (Zhao et al, *Cell Res.* 25, 1219-36, 2015). Taken together with our DSF results showing that only the biologically active SL analogs induce the melting temperature shift of D14, these data suggest that D14 destabilization correlates with D3 binding. We have added explanation about these reported data to our revised manuscript (line 201-204). We apologize for our poor explanation in the Method section, which has been edited in our revised manuscript (line 547-560).

4. Endogenous and synthetic SL can have very different chemical scaffolds. Extensive structure-function analysis has identified as minimum requirement the SL D-ring moiety attached by a labile, hydrolyzable bond to a variety of scaffolds. Also, limited previous attempts to mutationally separate D14 SL-responsiveness from SL hydrolysis had been unsuccessful, in part because residues important for SL catalysis have also been important for SL binding. Here the authors provide evidence that an alanine mutation of the catalytic triad residue D218 (which does not directly bind SL) severely compromises SL hydrolysis, as expected, yet as an overexpressed protein is still capable to complement a d14 mutation and to interact with SMXL7 in a Y2H assay. SMXL7 and D3 can both independently bind to D14 at supposedly two different surfaces, but can also directly bind each other. To gain mechanistic insight, it would therefore be

important to see whether D14 D218A retains the ability to interact with D3 in a SL-dependent manner (Y2H or other assay).

→Response

Thank you so much for an important suggestion. We have done a Y3H assay using AtD14, MAX2 and ASK1 as a third protein in the revised manuscript (Supplementary Fig. 6d, line 328-333, line 659-677). In the presence of the third protein, ASK1, we could successfully detect the interaction between AtD14^{D218A} and MAX2. Thus, now our data clearly show that the D218A mutant is able to interact with both partner proteins, SMXL7 and MAX2, in an SL-dependent manner.

5. The authors identified a new d14 mutant, R233H (d14-2), located in the helical lid domain. D14 R233H is catalytically active, but failed to complement d14 and has a weakened Y2H interaction with SMXL7. Overexpression of D14 R233H in wildtype cells resulted in increased branching, suggesting that it retains the ability to hydrolyze endogenous SL without inducing SL signaling. The authors interpret this phenotype as support for their hypothesis that SL hydrolysis is a deactivating step. Mutation of the same residue (R183E in their aa labelling) has previously been demonstrated to disrupt D3 binding (Zhao et al, 2015), while several groups have shown that the SL D-ring alone has very low D14 binding affinity and would not bind D14 under conditions. The mutational phenotype can therefore simply be explained by the inability of the D14 R233 mutant to interact with D3 (and therefore to signal), while it remains active in hydrolyzing endogenous SL which would prevent their ability to bind D14 and form covalently bound reaction intermediates (CLIM).

→Response

Thank you for a valuable comment. As the reviewer pointed out, the data with D14-2 type mutant alone are not sufficient to discuss about the catalytic function of D14. Here, we summarize some important data including published results.

1. As the reviewer commented, the hydrolysis products neither interact with D14 *in vitro* nor inhibit shoot branching *in vivo* (our data; Hamiaux et al., *Curr. Biol.* 22, 2032-6, 2012; Nakamura et al, *Nat. Commun.* 4, 2613, 2013).

2. The first half of our paper demonstrated that intact SL, but not CLIM, has a critical role to trigger the D14 active signaling state. This suggests that the SL molecules are not hydrolyzed when the signal is transduced, and the hydrolysis should occur after this signaling process.

3. The transgenic plants expressing enzymatically inactive AtD14^{D218A} mutant in the *atd14 max4* double mutant background becomes hypersensitive to exogenously applied SL, possibly because of the defect in hydrolytic degradation of bioactive SLs.

4. Overexpression of OsD14^{R233H}/AtD14^{R183H} in WT background resulted in a decrease in the endogenous SL levels and increased shoot branching.

Considering all together, these results suggest that the hydrolytic degradation occurs after the signal transduction to decompose the bioactive SL molecules. As the reviewer pointed out, our explanation in the original manuscript was insufficient to explain these things, thus we have revised our manuscript carefully (line 355-356, 387-390, 399-415).

In summary, the manuscript provides unexpected and important new insight indicating that the current model of SL signaling is incomplete, yet falls short in providing the evidence for the proposed alternative model.

Reviewer #3 (Remarks to the Author):

This manuscript described biochemical and genetic experiments on the detailed mechanism of strigolactone (SL) signalling by the D14 receptor. The authors argue that their work overturns the current model of SL signalling, and that SL signalling does not require hydrolysis of SL molecules.

My overall impression is that this is very important data, which needs to be published. However, I feel that the preparation of the manuscript is little bit rushed and incoherent. There is a huge of data here (especially in supplementary figures), but I'm not sure it has been put together in a particularly logical way. Some experiments are not at all well described, and just appear 'out of the blue' within the manuscript.

I have a few major suggestions for improving the manuscript.

1) In the first section of the manuscript (e.g. Figure 2) strongly suggest that SL cleavage is not needed for signalling, I do not think the data are completely unambiguous, because the level of signalling still correlates with the ability of the protein to cleave SLs. So when there is no more GR24 to cleave, the signalling stops. In CN-PMF, the substrate is still cleaved, at a slower rate, and signalling is less. There it still substrate at 6 hours, and so signalling continues. The data here do not completely rule out cleavage being necessary for signalling. While I find the data are convincing, I think the authors need to be slightly careful about the strength of their interpretation from these experiments. For instance, line 231: "our data clearly demonstrate that the uncleaved substrate itself not the hydrolysis intermediate or the products, induces the T_m shift of AtD14". To me, the authors data do suggest this, but I disagree that they 'clearly' demonstrate this.

Furthermore, the authors say that they have an completely undegradable SL

(3,6'-dihydroGR24), but they say that is biologically inactive. This suggests that cleavage IS necessary for signalling. If the authors' model is correct 3,6'-dihydroGR24 should still trigger signalling. So the authors should need to show whether 3,6'-dihydroGR24 triggers signalling by DSF. The only other explanation is that 3,6'-dihydroGR24 does not bind D14 at all - in which case it is not surprising it is neither biologically inactive nor cleaved by D14 - but which would make it completely irrelevant to the paper. Either way, this issue needs resolving.

→Response

We thank the reviewer for an important comment to improve our manuscript. As the reviewer suggested, our interpretation was partly too strong. We have revised our manuscript according to this suggestion (line 254).

As for 3,6'-dihydroGR24, it could not lower the melting temperature of AtD14 (Fig 2a), suggesting that this analog does not have an ability to interact with AtD14. As the reviewer pointed out, this is not a surprising result, but we are showing this result just to emphasize the fact that biologically inactive analog cannot induce the melting temperature shift.

2) In the second section of the manuscript, the authors present evidence that D14 is NOT covalently modified by GR24, and remains available to act up on more GR24. This is based on D14 being able to metabolize all GR24 despite there being a large molar excess of GR24. I feel that the experiment presented in Fig 2B could be made more convincing if the authors 'spike in' fresh GR24 after 4 hours, and show that the original D14 protein is still able to act up on this fresh ligand/substrate.

→Response

Thank you so much for a very valuable idea to improve our manuscript. According to the comment, we have performed the experiments using pre-incubated AtD14 protein with GR24 (Supplementary Fig. 5, line 293-299). As a result, we found that the AtD14 protein was still capable to hydrolyze freshly added GR24 after consuming the initially added substrate (4 h). Moreover, a clear melting temperature shift was detected with this sample. These data further support the notion that AtD14 is not a single-turn-over enzyme with GR24, and suggest that the transition of the AtD14 active state depends on intact SL.

3) In the third section of the manuscript, I found it quite strange that some catalytic triad mutants could still partially rescue d14, but not all of them. Clearly, the ability of the D218A mutant to rescue d14 supports the authors' model that cleavage is not necessary for signalling. I don't dispute the data, but the authors do not provide a convincing explanation for this observation for the differences between the catalytic mutants, which I find weakens their argument.

There is only a statement that "Therefore, the mutation to Ser and His possibly affected

the initial interaction with SLs." I would find the manuscript more convincing if the authors could provide some evidence data (perhaps from modelling protein structure) to show that this explanation (i.e. that some catalytic triad mutants don't bind SLs, let alone cleave them) is correct.

Again, I disagree with the statement at line 290:

"These results, together with the time-course DSF experiments, conclusively demonstrate that the hydrolase reaction catalyzed by D14 is not necessary for the signal transducing role." I don't think the data is completely conclusive, because there are still catalytic mutants that cannot perform signalling, for which the authors have not provided a convincing explanation.

→Response

Thank you for a critical suggestion. According to the comments, we have performed direct binding assays using the catalytic triad mutant of AtD14 (Supplementary Fig. 7c, line 342-345, 700-710). We found that D218A and S97C mutant proteins, which were capable to complement the *atd14* mutant, had stronger binding activity with SL than the other mutant proteins (Supplementary Fig. 7c). We have also prepared a figure explaining our idea more clearly, in which we show that the spatial position between the active site pocket and the three catalytic triad amino acids (Supplementary Fig. 7b, line 339).

We agree that our statement was too strong for our data, thus we changed 'conclusively demonstrate' to 'strongly suggest' (line 360).

4) In the fourth section of the manuscript, the structural biology work appears completely out of the blue without any explanation. The authors simply say: "When we reanalyzed the structural changes in AtD14".

There are no details in the M&M as to how the authors did this. Nor do they provide any in the results section. It is completely unclear whether this is an in-silico reassessment of structure, or if they have recrystallized the protein. How did the authors do this? Why did they do it? Why do they find different results to previously published. We need to know all these things.

Also, I think these data are primary data and should be in a main figure, because they would represent a major change in our understanding of D14 structure and function.

→Response

We apologize that we didn't explain the detail about the structural analysis of the conformationally altered AtD14. We didn't crystalize the protein by ourselves. We just downloaded the published pdb data (code; 5HZG) from a protein data bank, and analyzed the structure with a PyMOL software. In the revised manuscript, we have

added the explanation in the results section, and we have made a new paragraph in the Methods section (line 373-374, line 711-715).

5) In the fifth section, the last part of the section is very poorly explained. I didn't follow the logic of this at all.

"We also found that the levels of an endogenous SL, 4-deoxyorobanchol (4DO), in the rice overexpressors were decreased relative to WT and the empty vector-expressing plants (Fig. 4e). If considering the quite weak or no biological activities with SL hydrolysis products our data suggest that the hydrolase reaction catalyzed by D14 would be a deactivating step of SLs after transducing the signal."

This needs rewriting or removing.

→Response

We apologize that the explanation of this section was poor. We have revised the manuscript to explain this part more clearly (line 385-416). And also please see the answer for the question #5 of the reviewer #2.

Minor points:

1) Line 105: I don't think we really do have a 'near complete picture' of SL synthesis. There is still lots unknown.

→Response

We have corrected this part according to the comment (line 104).

2) Line 116: D53/SMXL7 is not 'a signalling suppressor' protein. The signalling pathways ends with degradation of D53, there is nothing for D53 to suppress. It might be a 'transcriptional repressor', but that is not the same as a 'signalling suppressor'.

→Response

We have changed the words, 'signalling suppressor', to 'negative regulator of SL signalling' (line 116-117, line 448).

3) Line 136: Gunilla et al is claimed as reference 21, but that is not what is stated in the reference list.

→Response

We were using the first name (Gunilla) of this author instead of the last name (Carlsonn). We have revised it (line 138).

4) Line 171: Br-PMF and CN-PMF are not introduced properly. Please state what they are.

→Response

According to the comment, we have added proper explanations about these chemicals (line 172-175).

5) Line 182: GR5 is not introduced at all. Please state what it is.

→Response

According to the comment, we have added a proper explanation about this chemical (line 187-189).

6) "the second attack from a water molecule must be occurred immediately after the first attack from"

This sentence appears out of nowhere, without any previous discussion of the chemical mechanism of D14 catalytic activity. It also makes no sense unless the reader has a background in chemistry. Please modify.

→Response

According to the comment, we have added proper explanations about the hydrolysis reaction mechanism catalyzed by this protein family (line 217-227).

7) Line 219 "denoting"  suggesting

→Response

We have edited according to the comment (line 241).

8) Line 275 "a little far from it". This is not a scientific statement. Define this better.

→Response

We have changed the statement to "doesn't contact with the pocket surface" (line 339).

9) Line 317: Given the manuscript already uses *Atd14-2*, it is quite confusing that the authors name their new rice mutant *d14-2*. To avoid confusion, can they please refer to this new mutant as *Osd14-2*, to make it completely clear what is being referred to.

→Response

According to the comments, we refer this new mutant allele from rice to *osd14-2* in this paper (line 391, 586, 593, 818).

10) Line 336. Can the authors perhaps summarise here the evidence that suggests D14 degrades SLs after signal transmission?

→Response

We have added an explanation about this question at the end of the previous section (line 411-415)

11) Line 339: Why is this model in a supplementary figure, it is clearly the main finding of the paper? Replace Figure 5 with this figure.

→Response

According to the comment, we have replaced Figure 5 with the model in the supplementary figure.

12) Line 355: Where is the evidence to this statement? The data presented in supplementary figure 10 only provide very circumstantial evidence to support this idea.

→Response

We agree with the reviewer's comment. According to the next comment, we have decided to remove this supplementary figure from our manuscript.

13) In my opinion, the authors should not be continuing to introduce data during the discussion. Either supplementary figure 10 is important enough to mention during the results section, or it is not that important. I don't think supplementary figure 10 helps the authors arguments, and should be cut.

→Response

Please see the answer for question #12.

14) Figure 3e. In the legend, the panels are described as 'upper' and 'lower', but they are next to each other.

→Response

Thank you for pointing out the error. We have edited according to the comment (Fig. 3e).

REVIEWERS' COMMENTS:

Reviewer #1 (Remarks to the Author):

The paper entitled Strigolactone perception and deactivation by a hydrolase receptor, DWARF14 by Seto, Yamaguchi et al. deals with the elucidation of the perception mechanism of Strigolactones, the recently discovered class of plant hormones.

In my first review of this paper I had already found the results convincing and in line with the hypothesis put forward by the authors. The additional experimental data the authors collected as a consequence of the other referees' remarks undoubtedly gave strength to the hypothesis that the hydrolysis of active SL-like compounds takes place after the perception and it is not in itself the event that induces downstream signaling. More in detail, the time course MS experiment showing the formation of a covalent D-ring/D14 complex and its half-life time, allows to contextualize the authors' new findings with the recent literature about His247 and D-ring covalent bond. I also appreciated the experiment of spiking fresh GR24 into the medium to show the activity of D14 in time.

However, in this second version, the text is conveying a less clear-cut message. This of course may depend on the attempt by the authors to meet the referees' criticism and to soften the first version. I would suggest the authors to be braver, at least in some critical points, as here below detailed:

Line 104: please remove "part", if it is true that the whole biosynthetic pathway is not completely elucidated, the points highlighted here (carotenoid derivatives and carlactone as key intermediate) are proven and accepted by the scientific community.

Line 196-204. Put it simpler and conclusive. T_m shift in DSF experiments directly correlates with signaling, hydrolysis not.

The mention to the behavior of 3,6'-dihydroGR24 can be misleading and confusing for the reader. Based on the results, active compounds induce T_m shift regardless their hydrolysability. As a consequence, you may have:

Active compounds, hydrolysable (classical SLs and analogues)

Active compounds, non-hydrolysable (or weakly hydrolysable, debranones)

Inactive compounds, hydrolysable or not hydrolysable, they are simply not perceived (3,6'-dihydroGR24).

Line 224, remove "junction". The activated water molecule triggers the second attack to the D14/D-ring complex to cleave the covalent bond, and HMB is released as the second product.

Line 217-232: this is a very sensitive paragraph and should be explained very clearly. In line 217 before "the hydrolysis reaction.." please add "our experimental results point at the hydrolysis reaction as a ..." or something similar...In this way, you make clear to the reader what is your hypothesis and why is different from the literature (line 225).

Abstract: I found the abstract sort of "shy"..not in line with the conclusions, these one very clear. Please add after the description of the rationale and the experiments of the paper, a phrase as "....in this paper we described how are intact SL molecules to induce the active signaling state of D14, and that D14 deactivates bioactive SLs by the hydrolytic degradation only after signal transmission.....or similar.

Reviewer #2 (Remarks to the Author):

The manuscript has been greatly improved and the additional experiments are very supportive of the proposed model. These data will be of great interest to the field. I therefore recommend the publication of the paper in its current form.

Reviewer #3 (Remarks to the Author):

The authors have addressed all my major concerns, and I feel that this is now a convincing and important piece of work.

The manuscript still needs a good copy-edit for both the quality of English and also for style. There are quite a lot of errors scattered through the manuscript.

These are some examples I picked up, but there are lots more in the manuscript that the authors need to correct.

Line 104 - "recent studies have revealed a part of how SLs are biosynthesized"
 "recent studies have revealed how SLs are synthesised"

Line 227 "which was speculated to be significantly slow"
 "which was speculated to be significantly slower"

Line 286 "Notably, an over-time increment"
 "Notably, a gradual increment"

Line 339 "doesn't contact with the pocket surface"
 "does not form part of the pocket surface"

Line 342 "In consistence with this"
 "consistent with this"

Line 388 " Then, what is the physiological role of SL hydrolysis by D14"
 "What then is the physiological role of SL hydrolysis by D14"

Line 439 "the disordering of the Asp loop induced the coming off of the SL molecule"
 "the disordering of the Asp loop induced the detachment of the SL molecule"

Supp Fig 5A - Legend says incubation time is 'hours', but graph axis says 'mins'

Response to the reviewers comments.

(To find the edited portions, please refer to the line number in the main text file without showing the track changes)

Reviewer #1 (Remarks to the Author):

The paper entitled Strigolactone perception and deactivation by a hydrolase receptor, DWARF14 by Seto, Yamaguchi et al. deals with the elucidation of the perception mechanism of Strigolactones, the recently discovered class of plant hormones.

In my first review of this paper I had already found the results convincing and in line with the hypothesis put forward by the authors. The additional experimental data the authors collected as a consequence of the other referees' remarks undoubtedly gave strengths to the hypothesis that the hydrolysis of active SL-like compounds takes place after the perception and it is not in itself the event that induces downstream signaling. More in details, the time course MS experiment showing the formation of a covalent D-ring/D14 complex and its half-life time, allows to contextualize the authors' new findings with the recent literature about His247 and D-ring covalent bond. I also appreciated the experiment of spiking fresh GR24 into the medium to show the activity of D14 in time.

However, in this second version, the text is conveying a less clear-cut message. This of course may depend on the attempt by the authors to meet the referees' criticism and to soften the first version. I would suggest the authors to be braver, at least in some critical points, as here below detailed:

Line 104: please remove "part", if it is true that the whole biosynthetic pathways is not completely elucidated, the points highlighted here (carotenoids derivatives and carlactone as key intermediate) are proven and accepted by the scientific community.

→We have edited our manuscript according to the comment (line 105 in the revised manuscript file).

Line 196-204. Put it simpler and conclusive. T_m shift in DSF experiments directly correlates with signaling, hydrolysis not.

The mention to the behavior of 3,6'-dihydroGR24 can be misleading and confusing for the reader. Based on the results, active compounds induce T_m shift regardless their hydrolysability. As a consequence, you may have:

Active compounds, hydrolysable (classical SLs and analogues)

Active compounds, non-hydrolysable (or weakly hydrolysable, debranones)

Inactive compounds, hydrolysable or not hydrolysable, they are simply not perceived (3,6'-dihydroGR24).

→We have edited our manuscript according to the comment. (line198-200)

Line 224, remove “junction”. The activated water molecule triggers the second attack to the D14/D-ring complex to cleave the covalent bond, and HMB is released as the second product.

→We have edited our manuscript according to the comment. (line 223 in the revised manuscript file)

Line 217-232: this is a very sensitive paragraph and should be explained very clearly. In line 217 before “the hydrolysis reaction..” please add “our experimental results point at the hydrolysis reaction as a ...” or something similar...In this way, you make clear to the reader what is your hypothesis and why is different from the literature (line 225).

→Thank you for a comment. Here we are explaining about the general hydrolysis reaction mechanism catalyzed by the α/β -hydrolase family proteins. Thus we added a word, ‘generally’ at the beginning of the sentence. (line 217 in the revised manuscript file)

Abstract: I found the abstract sort of “shy”..not in line with the conclusions, these one very clear.

Please add after the description of the rationale and the experiments of the paper, a phrase as “...in this paper we described how are intact SL molecules to induce the active signaling state of D14, and that D14 deactivates bioactive SLs by the hydrolytic degradation only after signal transmission.....or similar.

→We have edited the abstract according to the comment.

Reviewer #2 (Remarks to the Author):

The manuscript has been greatly improved and the additional experiments are very supportive of the proposed model. These data will be of great interest to the field. I therefore recommend the publication of the paper in its current form.

Reviewer #3 (Remarks to the Author):

The authors have addressed all my major concerns, and I feel that this is now a convincing and important piece of work.

The manuscript still needs a good copy-edit for both the quality of English and also for style. There are quite a lot of errors scattered through the manuscript.

These are some examples I picked up, but there are lots more in the manuscript that the authors need to correct.

Line 104 - "recent studies have revealed a part of how SLs are biosynthesized"

 "recent studies have revealed how SLs are synthesised"

→We have edited our manuscript according to the comment. (line 105 in the revised manuscript file)

Line 227 "which was speculated to be significantly slow"

 "which was speculated to be significantly slower"

→We have edited our manuscript according to the comment. (line 227 in the revised manuscript file)

Line 286 "Notably, an over-time increment"

 "Notably, a gradual increment"

→We have edited our manuscript according to the comment. (line 286 in the revised manuscript file)

Line 339 "doesn't contact with the pocket surface"

 "does not form part of the pocket surface"

→We have edited our manuscript according to the comment. (line 339 in the revised manuscript file)

Line 342 "In consistence with this"

 "consistent with this"

→We have edited our manuscript according to the comment. (line 342 in the revised manuscript file)

Line 388 " Then, what is the physiological role of SL hydrolysis by D14"

 "What then is the physiological role of SL hydrolysis by D14"

→We have edited our manuscript according to the comment. (line 387 in the revised manuscript file)

Line 439 "the disordering of the Asp loop induced the coming off of the SL molecule"

 "the disordering of the Asp loop induced the detachment of the SL molecule"

→We have edited our manuscript according to the comment. (line 438 in the revised manuscript file)

Supp Fig 5A - Legend says incubation time is 'hours', but graph axis says 'mins'

→We have edited our manuscript according to the comment.